# The Label Horizon Paradox: Rethinking Supervision Targets in Financial Forecasting

**Chen-Hui Song** [1]  **Shuoling Liu** [1†]  **Liyuan Chen** [1†]

## Abstract

While deep learning has revolutionized financial forecasting through sophisticated architectures, the design of the supervision signal itself is rarely scrutinized. We challenge the canonical assumption that training labels must strictly mirror inference targets, uncovering the **Label Horizon Paradox**: the optimal supervision signal often deviates from the prediction goal, shifting across intermediate horizons governed by market dynamics. We theoretically ground this phenomenon in a dynamic signal-noise trade-off, demonstrating that generalization hinges on the competition between marginal signal realization and noise accumulation. To operationalize this insight, we propose a bi-level optimization framework that autonomously identifies the optimal proxy label within a single training run. Extensive experiments on large-scale financial datasets demonstrate consistent improvements over conventional baselines, thereby opening new avenues for **label-centric** research in financial forecasting.

## 1. Introduction

Deep learning has fundamentally transformed the landscape of quantitative finance, serving as a critical tool for high-noise time-series forecasting, particularly in short-term stock prediction (Al-Khasawneh et al., 2025; Chen et al., 2025; Sonkavde et al., 2023; Shah et al., 2022). The primary objective in this domain is to forecast the relative returns of assets to construct profitable portfolios. Unlike typical tasks in computer vision or natural language processing, financial prediction operates in an environment characterized by extremely low signal-to-noise ratios and non-stationary

dynamics. To tackle these intrinsic challenges, the research community has traditionally divided its efforts into two main streams: **Data-centric** approaches (Shi et al., 2025; Yu et al., 2023; Sawhney et al., 2020), which focus on engineering expressive alpha factors from limit order books and alternative data; and **Model-centric** approaches (Liu et al., 2024a; Wang et al., 2025; Liu et al., 2025; Chen & Wang, 2025), which design sophisticated architectures—ranging from RNNs to Transformers—to capture complex temporal dependencies.

However, this extensive focus on input representations and model architectures has left a critical component largely unexamined: the prediction target (or label) itself. In the standard paradigm, the training label is strictly aligned with the inference goal. For instance, in a daily prediction task (where predictions are made at time $t$ for the horizon $t + \Delta$, with $\Delta = 1$ day), it is typically taken for granted that the model must be trained on realized next-day returns. This convention implicitly assumes that bringing the supervision signal closer to the evaluation target is always beneficial, rarely questioning the optimality of the label's time horizon.

This raises a fundamental question: *Is the "correct" inference target necessarily the best training signal?* In this paper, we answer this in the negative. Through extensive experiments, we uncover a counter-intuitive phenomenon we term the **Label Horizon Paradox**:

*Minimizing training error on the canonical target horizon $t + \Delta$ does not guarantee optimal generalization on $t + \Delta$. Contrary to intuition, the most effective supervision signal is often **misaligned** with the inference target, residing at an intermediate horizon $t + \delta$ (where $\delta \neq \Delta$) that better balances signal realization against noise.*

Underlying this paradox is a fundamental trade-off governed by the temporal evolution of market information. We conceptualize generalization performance not as a static property, but as the outcome of a dynamic interplay between two competing rates:

**1. Marginal Signal Realization (Information Gain):** Information (Alpha) requires time to be absorbed by the market and fully priced in (Hong & Stein, 1999; Shleifer, 2000).

---

[†]Project Lead [1]E Fund Management Co., Ltd., Guangzhou, Guangdong, China. Correspondence to: Chen-Hui Song <songchenhui@efunds.com.cn>, Shuoling Liu <liushuoling@efunds.com.cn>, Liyuan Chen <chenly@efunds.com.cn>.

*Proceedings of the 43$^{rd}$ International Conference on Machine Learning*, Seoul, South Korea. PMLR 306, 2026. Copyright 2026 by the author(s).

**2. Marginal Noise Accumulation (Noise Penalty):** Simultaneously, as the time window expands, the market accumulates idiosyncratic volatility and stochastic shocks (Ang et al., 2006; Jiang et al., 2009) unrelated to the initial signal.

The generalization behavior is therefore governed by the interplay between these two rates. Extending the label horizon is beneficial only when marginal signal realization outpaces noise accumulation. Conversely, once the signal is largely priced in, the diminishing information gain is overwhelmed by compounding noise, rendering further extension detrimental. The optimal horizon $\delta^*$ therefore emerges at the precise equilibrium where marginal information gain equals the marginal noise penalty.

Crucially, since the underlying rates of signal realization and noise accumulation are unknown and dynamic, the optimal horizon $\delta^*$ cannot be hard-coded a priori. To operationalize this insight, we apply a Bi-level Optimization Framework (Chen et al., 2022; Franceschi et al., 2018) for adaptive horizon learning. Instead of manually selecting a fixed proxy, our method treats the label horizon as a learnable parameter. By formulating the problem as a bi-level objective, the model automatically learns to weight different horizons, dynamically discovering the sweet spot where this trade-off is maximized for the specific dataset and model architecture.

In this work, we focus on short-term stock forecasting and make the following primary contributions:

**1. Theoretical Unification:** Grounded in Arbitrage Pricing Theory, we provide a rigorous derivation using a linear factor model to quantify the temporal dynamics of signal realization and noise accumulation. This theoretical foundation unifies previously fragmented empirical observations, formally establishing the signal-noise trade-off as the primary driver of model generalization in financial forecasting.

**2. Methodological Innovation:** Motivated by our theoretical insights, we propose a novel, end-to-end adaptive framework. By formulating temporal-horizon selection as a dynamic optimization problem, our method automatically identifies the optimal supervision signal in a single training run, eliminating the need for the computationally prohibitive brute-force search required by traditional methods.

**3. Empirical Validation:** We extensively evaluate our framework across ten diverse architectures. Our approach consistently yields significant predictive improvements in the emerging A-share market (CSI 300, 500, and 1000) and demonstrates robust cross-market generalizability in the highly efficient US equity market (S&P 500). Furthermore, downstream backtesting and severe macroeconomic stress testing (e.g., on the 2024 market crash) confirm its exceptional resilience and practical profitability.

## 2. Preliminaries

In this section, we formalize the short-term stock cross-sectional prediction task (Linnainmaa & Roberts, 2018) and outline the deep learning framework employed.

### 2.1. Stock Cross-Sectional Prediction

Consider a universe of $N$ stocks at a decision time $t$. Our goal is to forecast the relative performance of these assets over a subsequent fixed period, denoted as the target horizon $\Delta$ (where $\Delta$ represents the number of time steps in minutes). Let $p_{i,t}$ denote the price of stock $i$ at time $t$. The target variable, the target realized return, is defined as $r_{i,t}^{\Delta} = p_{i,t+\Delta}/p_{i,t} - 1$.

We denote the simultaneous returns of the entire market as a cross-sectional vector $\mathbf{r}_t^{\Delta} = [r_{1,t}^{\Delta}, \ldots, r_{N,t}^{\Delta}]^{\top} \in \mathbb{R}^N$.

### 2.2. Optimization Objective

In quantitative investment, the primary goal is to construct a portfolio that maximizes risk-adjusted returns. This objective is theoretically grounded in the Fundamental Law of Active Management (Grinold & Kahn, 2000), which relates the expected Information Ratio (IR) of a strategy to its predictive power:

$$\mathbb{E}[\text{IR}] \approx \text{IC} \cdot \sqrt{\text{Breadth}}. \qquad (1)$$

Intuitively, this law asserts that performance is driven by the quality of predictions and the number of independent trading opportunities. Specifically, Breadth represents the number of independent bets (proportional to the universe size $N$), and IC (Information Coefficient) is the Pearson correlation coefficient $\rho$ between the predicted scores and the realized return vector $\mathbf{r}_t^{\Delta}$.

Since market breadth is generally fixed for a given strategy, maximizing portfolio performance is mathematically equivalent to maximizing the IC. Consequently, our learning objective is to train a model that produces scores maximally correlated with the target return $\mathbf{r}_t^{\Delta}$. In the sequel, we use IC and $\rho$ interchangeably to denote this correlation.

### 2.3. Deep Learning Framework

Modern deep learning approaches capture market dynamics by modeling stock features as time series. For each stock $i$ at time $t$, the input is a sequence of historical feature vectors $\mathbf{x}_{i,t} \in \mathbb{R}^{L \times D}$, spanning a lookback window of size $L$ with $D$ feature channels. Aggregating across the universe, the input at time $t$ forms a 3-dimensional tensor $\mathbf{X}_t \in \mathbb{R}^{N \times L \times D}$.

A deep neural network $f_\theta$ (e.g., LSTM, GRU, or Transformer) encodes this history into a predictive score:

$$\hat{\mathbf{y}}_t = f_\theta(\mathbf{X}_t) \in \mathbb{R}^N. \qquad (2)$$

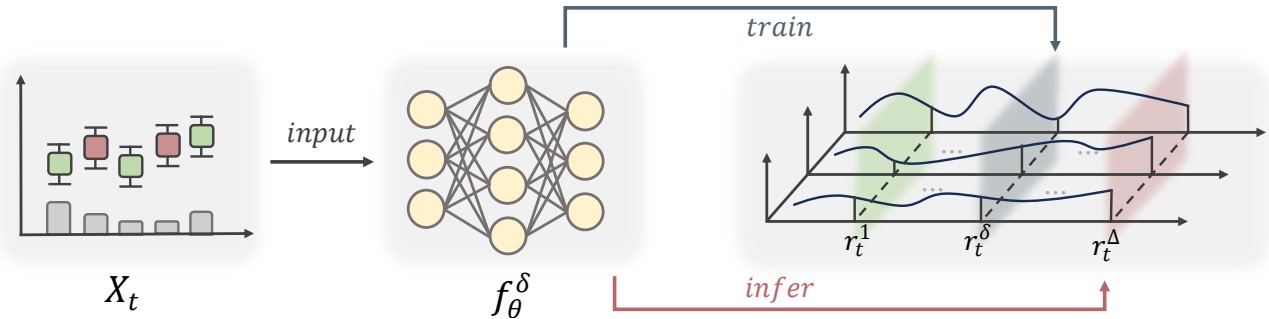

*Figure 1.* **Training and Inference Pipeline.** The leftmost panel shows historical input features $X_t$, which are processed by a neural network model $f_\theta^\delta$. The right panel illustrates unfolding future price paths with different time horizons $(r_t^1, r_t^\delta, r_t^\Delta)$. The arrows highlight a central premise of this study: during training (top arrow), the model may be optimized against an intermediate proxy label $r_t^\delta$; however, during inference (bottom arrow), the model's performance is strictly evaluated on its ability to forecast the final target return ($r_t^\Delta$). Our goal is not to change the evaluation target, but to question and improve the choice of training label that best serves this fixed objective.

Standard financial forecasting paradigms rigidly align the supervision label with the final inference goal. Typically, models are trained to minimize a loss $\mathcal{L}(\theta) = \sum_t \ell(\hat{\mathbf{y}}_t, \mathbf{y}_t)$ where the label $\mathbf{y}_t$ is set strictly to the target return $\mathbf{r}_t^\Delta$. This convention implicitly assumes that the target horizon provides the most effective learning signal, thereby defaulting to the label horizon that conceptually matches the evaluation metric.

However, relying solely on the terminal snapshot at $t + \Delta$ ignores the continuous price discovery process leading up to that point. To investigate whether the trajectory offers better supervision, we introduce a granular notation for intermediate dynamics. Let $\delta \in \{1, \ldots, \Delta\}$ denote the discretized time index within the prediction window. We define $p_{i,t+\delta}$ as the price of stock $i$ at step $\delta$. Consequently, the cumulative return from decision time $t$ to this intermediate horizon is formulated as $r_{i,t}^\delta = p_{i,t+\delta}/p_{i,t} - 1$.

Aggregating these across the universe yields the intermediate return vector $\mathbf{r}_t^\delta \in \mathbb{R}^N$. In this work, we challenge the dogma that the optimal supervision signal must mirror the inference target (i.e., $\delta = \Delta$). Instead, we explore how utilizing these proxy vectors $\mathbf{r}_t^\delta$ as training labels can effectively enhance generalization on the final target $\mathbf{r}_t^\Delta$.

## 3. The Label Horizon Paradox

Contrary to the standard practice of strictly aligning training labels with prediction targets, the aforementioned Label Horizon Paradox reveals the limitations of this convention. This section presents empirical evidence of this phenomenon and introduces a theoretical mechanism to explain it.

### 3.1. Empirical Observation

To systematically evaluate the impact of label horizon on forecasting performance, we conducted a control experiment as illustrated in Figure 1. While our ultimate inference

goal remains fixed—forecasting stocks based on the realized return $\mathbf{r}^\Delta$ at the target horizon—we vary the supervision signal used during training. Specifically, we train a set of identical deep neural networks $\{f_\theta^{(\delta)}\}_\delta$, where each model is supervised by the cumulative return $\mathbf{r}_t^\delta$ at a specific intermediate horizon $\delta \in \{1, \ldots, \Delta\}$. For clarity, we instantiate this analysis using an LSTM backbone on the CSI 500 universe, as this mid-cap index offers a representative balance between liquidity and cross-sectional breadth, and LSTMs remain a strong and widely used baseline in short-term stock forecasting. Detailed experimental settings are provided in the Appendix A.

We examine this phenomenon across three distinct market scenarios, which represent different prediction horizons in quantitative finance:

**Scenario 1: Interday (Standard) Prediction.** The decision time $t$ is the market close on day $D$, and the prediction target $\mathbf{r}^\Delta$ is the return from the close of day $D$ to the close of day $D + 1$. Intermediate horizons correspond to cross-sectional returns at each minute of the next day. This setup follows the standard convention in stock forecasting studies.

**Scenario 2: Intraday (30-minute) Prediction.** The decision time $t$ is set to the exact midpoint of the daily trading session. The target horizon is a short-term interval of $\Delta = 30$ minutes immediately following this timestamp. The objective is to predict the return realized exclusively within this 30-minute window.

**Scenario 3: Intraday (90-minute) Prediction.** Similarly, the decision time $t$ is fixed at the midpoint of the trading session. The target horizon extends to $\Delta = 90$ minutes. The model aims to forecast the cumulative return realized over this longer intraday interval within the same session.

We define the optimal training horizon as $\delta^* = \text{argmax}_\delta \, \rho(\hat{\mathbf{y}}_t^\delta, \mathbf{r}_t^\Delta)$. As shown in Figure 2, the performance curves exhibit distinct patterns across these scenarios:

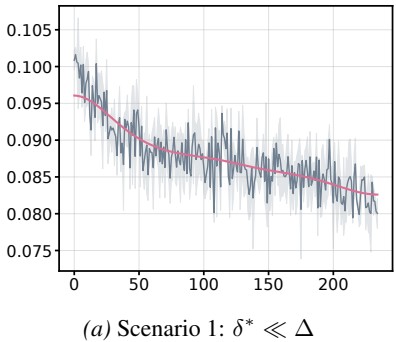

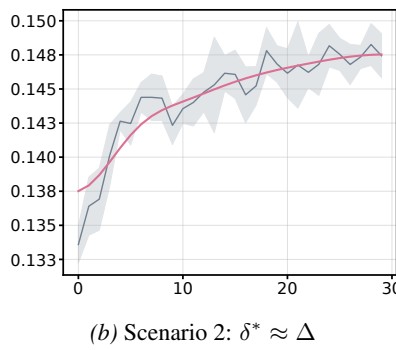

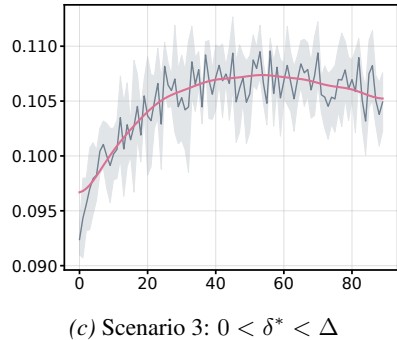

*(a)* Scenario 1: $\delta^* \ll \Delta$        *(b)* Scenario 2: $\delta^* \approx \Delta$        *(c)* Scenario 3: $0 < \delta^* < \Delta$

*Figure 2.* **Performance Curves across Different Scenarios.** The x-axis is the training horizon $\delta$, and the y-axis is the out-of-sample IC on the fixed final target $\mathbf{r}^\Delta$. The curves are obtained by training LSTM models on the CSI 500 dataset, with 5 independent models per horizon. The blue line shows the raw results, and the red line shows the Gaussian-smoothed trend.

**In Scenario 1**, we observe a **Monotonic Decrease**. Performance peaks at a very early horizon ($\delta^* \ll \Delta$) and degrades significantly as $\delta$ approaches $\Delta$. This strictly violates the closer-is-better intuition.

**In Scenario 2**, we observe a **Monotonic Increase**. Here, training on the target itself is optimal ($\delta^* \approx \Delta$). The signal realization dominates the noise accumulation, meaning the information gain continues until the target time. Consequently, training on the target itself is optimal.

**In Scenario 3**, the curve is typically **Hump-Shaped**. The optimal horizon lies at an intermediate point ($0 < \delta^* < \Delta$). This reflects an intraday trade-off: the model benefits from alpha realization but suffers when forced to fit unpredictable late-session noise.

These observations collectively confirm the Label Horizon Paradox: *Supervising the model with the exact target might yield suboptimal generalization*

### 3.2. Theoretical Framework

To provide a rigorous mechanism for the observations above, we analyze the problem through a Linear Factor Model grounded in **A**rbitrage **P**ricing **T**heory (Reinganum, 1981), hereafter referred to as APT. The details of our theoretical analysis are provided in Appendix C.

We begin by modeling the intrinsic dynamics of market returns.

**Assumption 3.1.** We extend the static APT framework to model the short-term cumulative return $r_{i,t}^\delta$ using observable factor exposures $\mathbf{s}_{i,t} \in \mathbb{R}^d$:

$$r_{i,t}^\delta = \alpha(\delta)\mathbf{w}^{*\top}\mathbf{s}_{i,t} + \epsilon_{i,t}^\delta, \quad \epsilon_{i,t}^\delta \sim \mathcal{N}(0, \sigma^2(\delta + \delta_0)) \quad (3)$$

Here, $\mathbf{w}^*$ denotes the factor risk premia vector (i.e., the ground-truth weight vector that linearly maps factor exposures to expected returns), $\alpha(\delta)$ represents the signal realization process (i.e., how much of the information has

been priced in by horizon $\delta$), and $\sigma^2$ denotes the rate of unpredictable noise accumulation.

Under this generative process, a model trained on the proxy horizon $\delta$ yields an estimator $\hat{\mathbf{w}}_\delta$ corrupted by the specific noise variance at that horizon. We define the generalization performance $J(\delta) := \rho^2(\hat{\mathbf{y}}_t^\delta, \mathbf{r}_t^\Delta)$ as the squared predictive correlation with the fixed target $\mathbf{r}_t^\Delta$. By decomposing the log-performance, we reveal the structural trade-off driving the paradox:

**Theorem 3.2.** *The expected performance is determined by the net balance of two competing accumulation processes (ignoring a constant term):*

$$\ln J(\delta) = \underbrace{2\ln\alpha(\delta)}_{\text{Information Gain}} - \underbrace{\ln\left[\alpha(\delta)^2 + K(\delta + \delta_0)\right]}_{\text{Noise Penalty}} \quad (4)$$

*where $K$ is a constant related to the model's estimation variance.*

Equation (4) quantifies the fundamental tension in labeling:

**1. Information Gain** reflects the growth of valid signal in the label. It increases as $\delta$ extends, but naturally saturates as $\alpha(\delta) \to const$ (when information is fully priced).

**2. Noise Penalty** reflects the growth of prediction uncertainty. Since the idiosyncratic variance $K(\delta + \delta_0)$ follows a random walk, this penalty grows strictly with time $\delta$.

Consequently, the optimal horizon $\delta^*$ defines the tipping point where the marginal accumulation of noise begins to outpace the marginal accumulation of information.

### 3.3. Generalization to Deep Learning

While Theorem 3.2 is derived for a linear estimator, its implications can be intuitively related to deep neural networks. From a representation-learning viewpoint, a deep model is often viewed as a non-linear feature extractor followed by a simple linear readout layer (Bengio et al., 2013; Alain

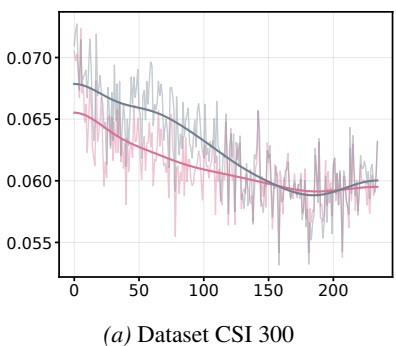 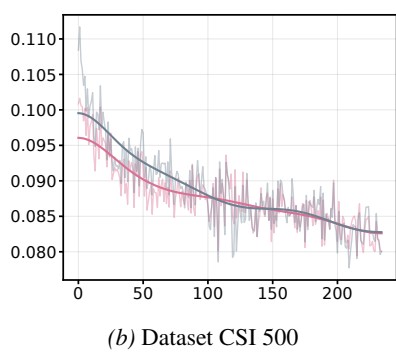 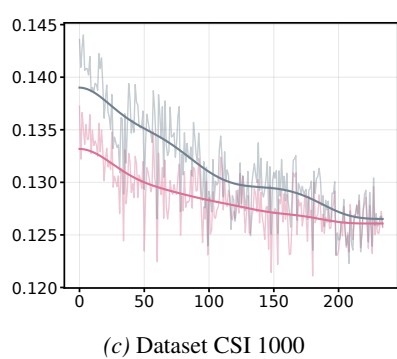

*(a) Dataset CSI 300*      *(b) Dataset CSI 500*      *(c) Dataset CSI 1000*

*Figure 3.* **Decomposition Validation.** The x-axis represents the training horizon $\delta$, and the y-axis represents the Out-of-Sample IC on the fixed final target $\mathbf{r}^{\Delta}$. We performed a dense experimental sweep to rigorously verify the theoretical decomposition, training distinct LSTM models for every minute-level horizon across 5 random seeds. The red lines depict the empirical Test IC on the inference target ($\Delta$), while the blue lines illustrate the theoretical values derived from the product term in Corollary 3.3. For both metrics, lighter shades correspond to raw measurements from individual trials, while solid darker curves indicate the Gaussian-smoothed trends.

& Bengio, 2016). The latent representation learned by the network can be thought of as playing the same role as the signal $\mathbf{s}_{i,t}$ in our framework, and the final linear layer then fits this signal under label noise, in line with standard linear or kernel-based generalization analyses (Belkin et al., 2019; Jacot et al., 2018). Under this perspective, a similar trade-off is expected to influence deep models as well: regardless of the complexity of the feature extractor, generalization is shaped by the balance between realized signal and accumulated noise in the supervision.

To empirically validate this theoretical connection, we specifically examine Scenario 1. Given the interday nature of this task, the market has ample time to absorb the prior day's information, causing the signal embedded in the input features to be priced in immediately upon the market open. Consequently, the signal realization saturates almost immediately ($\alpha(\delta) \approx const$). This leads to the following approximation:

**Corollary 3.3.** *Under the condition where signal evolution is static relative to noise accumulation, the final predictive correlation can be approximated as the product of two observable terms:*

$$\rho(\hat{\mathbf{y}}_t^{\delta}, \mathbf{r}_t^{\Delta}) \approx \rho(\hat{\mathbf{y}}_t^{\delta}, \mathbf{r}_t^{\delta}) \times \rho(\mathbf{r}_t^{\delta}, \mathbf{r}_t^{\Delta}) \qquad (5)$$

The detailed proof is provided in Appendix C.6.

We substantiate this corollary through large-scale experiments, as shown in Figure 3. Notably, the theoretical curve, calculated solely from the product on the right-hand side of Eq. (5), closely matches the actual performance with high precision across all datasets. This strong alignment confirms that the proposed signal-noise mechanism is indeed the driving force in deep learning models.

*Remark* 3.4. This decomposition can also be understood from a purely statistical perspective via the Partial Correlation Formula (Kenett et al., 2015). We discuss this alterna-

tive derivation in Appendix D, which further corroborates the robustness of our theory.

### 3.4. Mechanism Analysis

Leveraging the signal-noise decomposition, we can now interpret the distinct performance patterns observed across the three scenarios:

**Scenario 1 (Interday):** Interday signals typically saturate early (e.g., at the market open). Extending $\delta$ beyond this point yields vanishing information gain ($\alpha' \approx 0$) while the noise penalty continues to grow linearly. Thus, the gradient is strictly negative, driving the optimal horizon to be much smaller than the target ($\delta^* \ll \Delta$).

**Scenario 2 (30-min):** In short, high-momentum windows, the market is in a state of active price discovery. The signal realization rate remains high enough to suppress the accumulation of noise. Consequently, the gradient remains positive, making the full horizon optimal ($\delta^* \approx \Delta$).

**Scenario 3 (90-min):** Here, the signal's effective lifespan is shorter than the target window. Initially, rapid information gain improves performance; however, once the signal saturates, the persistent noise penalty eventually dominates, resulting in a peak at an intermediate horizon ($0 < \delta^* < \Delta$).

A detailed mathematical discussion, categorizing these regimes based on the sign of the derivative $d \ln J(\delta)/d\delta$, is provided in Appendix C.5.

## 4. Adaptive Horizon Learning via Bi-Level Optimization

Since the theoretically optimal horizon is dynamic and a brute-force search is expensive, we propose an automated Bi-level Optimization (BLO) framework. This method autonomously learns the ideal supervision during training,

stabilized by a warm-up phase and entropy regularization.

### 4.1. Bi-Level Optimization Framework

Our objective is to train a model $f_\theta$ using a set of candidate labels such that it generalizes best on the ultimate target horizon $\Delta$. We introduce a learnable weight vector $\boldsymbol{\lambda} \in \mathbb{R}^\Delta$ (obtained by applying a softmax to learnable logits, so $\sum \lambda_\delta = 1$) to govern the importance of each candidate horizon $\delta$. We treat $\boldsymbol{\lambda}$ as learnable parameters and optimize them via an intra-batch splitting strategy. In each iteration, a mini-batch $\mathcal{B}$ is split into a support set $\mathcal{B}_{in}$ and a query set $\mathcal{B}_{out}$.

**1. Inner Loop (Proxy Learning on $\mathcal{B}_{in}$).** The model parameters $\theta$ are updated on $\mathcal{B}_{in}$ with loss function $\ell$. The final objective $\mathcal{L}_{inner}$ is a weighted combination of losses against the candidate return labels $\mathbf{R}_t = \{\mathbf{r}_t^{(\delta)}\}_\delta$:

$$\mathcal{L}_{inner}(\theta, \boldsymbol{\lambda}) = \sum_{(\mathbf{X}_t, \mathbf{R}_t) \in \mathcal{B}_{in}} \sum_{\delta=1}^{\Delta} \lambda_\delta \cdot \ell(f_\theta(\mathbf{X}_t), \mathbf{r}_t^\delta) \quad (6)$$

Here, the model is guided by the composite signal emphasized by $\boldsymbol{\lambda}$.

**2. Outer Loop (Target Validation on $\mathcal{B}_{out}$).** The quality of the learned weights $\boldsymbol{\lambda}$ is verified on $\mathcal{B}_{out}$. Crucially, the outer loss consists of the validation error against the target Horizon $\mathbf{r}_t^\Delta$ and an entropy regularization term:

$$\min_{\boldsymbol{\lambda}} \quad \mathcal{L}_{outer}(\theta^*(\boldsymbol{\lambda}), \mathcal{B}_{out}) - \gamma H(\boldsymbol{\lambda}) \quad (7)$$

$$\text{s.t.} \quad \theta^*(\boldsymbol{\lambda}) = \arg\min_\theta \mathcal{L}_{inner}(\theta, \boldsymbol{\lambda}, \mathcal{B}_{in}) \quad (8)$$

$$\text{where} \quad H(\boldsymbol{\lambda}) = -\sum_\delta \lambda_\delta \log \lambda_\delta \quad (9)$$

$$\mathcal{L}_{outer} = \sum_{(\mathbf{X}_t, \mathbf{r}_t^\Delta) \in \mathcal{B}_{out}} \ell(f_{\theta^*}(\mathbf{X}_t), \mathbf{r}_t^\Delta) \quad (10)$$

The added entropy term $H(\boldsymbol{\lambda})$ acts as a safeguard against noise disturbance. It prevents the weight distribution from collapsing onto a single horizon—a scenario where transient noise artifacts could disproportionately dominate the gradient and lead to training instability. Instead, it maintains a smoother distribution, encouraging the model to leverage complementary information from multiple horizons.

### 4.2. Optimization Procedure

Solving the bi-level objective requires differentiating through the optimization path. We employ a two-phase strategy to ensure stability.

**Phase 1: Warm-up with Standardized Mean-Field.** Initiating the bi-level optimization directly from scratch is suboptimal, as the model parameters $\theta$ initially lack effective feature representations, making the meta-optimization landscape highly volatile and prone to training collapse.

To address this, we employ a warm-up phase using standard supervised learning. We construct a robust supervision signal by aggregating information across all horizons. However, since raw returns exhibit varying volatilities (scales), we first apply cross-sectional standardization to narrow the distributional gaps between different labels:

$$\mathbf{z}_t^\delta = \frac{\mathbf{r}_t^\delta - \mu_t^\delta}{\sigma_t^\delta} \quad (11)$$

For the first $N_{warm}$ epochs, the model is trained to minimize a single loss function $\ell$ against the arithmetic mean of these standardized candidates:

$$\bar{\mathbf{y}}_t = \frac{1}{\Delta} \sum_{\delta=1}^{\Delta} \mathbf{z}_t^\delta \quad (12)$$

This mean-field initialization efficiently establishes a foundational representation, ensuring a stable starting point for the subsequent bi-level adaptation.

**Phase 2: Bi-Level Update.** After warm-up, we enable the adaptive weighting scheme. Given a batch split $(\mathcal{B}_{in}, \mathcal{B}_{out})$:

**1. Inner Loop Update.** We simulate the learning trajectory by performing $M$ steps of gradient descent on $\mathcal{B}_{in}$. Starting from the current parameters $\theta$, the model is updated sequentially ($m = 1, \ldots, M$):

$$\theta_m(\boldsymbol{\lambda}) \leftarrow \theta_{m-1} - \eta \nabla_\theta \mathcal{L}_{inner}(\theta_{m-1}, \boldsymbol{\lambda}) \quad (13)$$

By retaining the computational graph of these updates, we derive the look-ahead state $\theta_M(\boldsymbol{\lambda})$, which is functionally dependent on $\boldsymbol{\lambda}$.

Crucially, since the model has already acquired a robust representation during the warm-up phase, a single gradient step ($M = 1$) is typically sufficient to capture the sensitivity of the parameters to the weights. This makes the process highly computationally efficient.

**2. Outer Loop Update.** We evaluate $\theta_M$ on $\mathcal{B}_{out}$ against the target $\mathbf{r}_t^\Delta$. We compute the gradient of the validation loss w.r.t. $\boldsymbol{\lambda}$, add the entropy gradient, and update $\boldsymbol{\lambda}$:

$$\boldsymbol{\lambda} \leftarrow \boldsymbol{\lambda} - \beta \nabla_{\boldsymbol{\lambda}} \left( \mathcal{L}_{outer}(\theta_M, \boldsymbol{\lambda}) - \gamma H(\boldsymbol{\lambda}) \right) \quad (14)$$

By iterating these steps, $\boldsymbol{\lambda}$ converges to a robust distribution that emphasizes the most effective horizons for supervision signal extraction, with $\delta^* = \arg\max_\delta \lambda_\delta$. Based on the selected optimal horizon, we then retrain the forecasting model using this horizon as the supervision target.

## 5. Experiments

In this section, we validate our proposed bi-level method through extensive experiments on real-world market data.

*Table 1.* **Main Results.** Comprehensive performance comparison between standard training (Std.) and our Bi-level framework (Ours) across three market indices. All results are averaged over 5 random seeds. Bold indicates the better performance.

| DATASET | MODEL | IC (×10) | | ICIR | | RANKIC (×10) | | RANKICIR | | TOP RET (%) | | SHARPE RATIO | |
|---|---|---|---|---|---|---|---|---|---|---|---|---|---|
| | | STD. | OURS | STD. | OURS | STD. | OURS | STD. | OURS | STD. | OURS | STD. | OURS |
| CSI 300 | LSTM | 0.637 | **0.720** | 0.443 | **0.562** | 0.669 | **0.727** | 0.520 | **0.556** | 0.231 | **0.240** | 2.332 | **2.390** |
| | GRU | 0.586 | **0.730** | 0.445 | **0.560** | 0.566 | **0.662** | 0.436 | **0.542** | 0.209 | **0.256** | 1.940 | **2.612** |
| | DLINEAR | 0.537 | **0.684** | 0.403 | **0.405** | 0.568 | **0.683** | 0.425 | **0.516** | 0.207 | **0.221** | 2.007 | **2.456** |
| | RLINEAR | 0.565 | **0.716** | 0.424 | **0.532** | 0.625 | **0.662** | 0.461 | **0.534** | 0.211 | **0.273** | 2.139 | **2.781** |
| | PATCHTST | 0.568 | **0.673** | 0.411 | **0.459** | 0.525 | **0.653** | 0.411 | **0.464** | 0.213 | **0.236** | 2.078 | **2.381** |
| | iTRANSFORMER | 0.540 | **0.644** | 0.413 | **0.421** | 0.560 | **0.650** | 0.428 | **0.496** | 0.216 | **0.230** | 2.196 | **2.518** |
| | MAMBA | 0.641 | **0.719** | 0.457 | **0.539** | 0.646 | **0.673** | 0.462 | **0.514** | 0.238 | **0.273** | 2.271 | **2.570** |
| | BI-MAMBA+ | 0.661 | **0.717** | 0.485 | **0.486** | 0.631 | **0.704** | 0.524 | **0.543** | 0.245 | **0.267** | 2.698 | **2.956** |
| | MODERNTCN | 0.565 | **0.686** | 0.394 | **0.423** | 0.571 | **0.694** | 0.448 | **0.548** | 0.218 | **0.227** | 2.390 | **2.498** |
| | TCN | 0.604 | **0.691** | 0.459 | **0.498** | 0.632 | **0.702** | 0.503 | **0.520** | 0.216 | **0.230** | 2.102 | **2.240** |
| CSI 500 | LSTM | 0.845 | **1.029** | 0.724 | **0.861** | 0.792 | **0.859** | 0.861 | **0.895** | 0.382 | **0.383** | 3.534 | **3.660** |
| | GRU | 0.862 | **1.079** | 0.752 | **0.886** | 0.812 | **0.883** | 0.846 | **0.939** | 0.359 | **0.382** | 3.398 | **3.458** |
| | DLINEAR | 0.839 | **0.986** | 0.632 | **0.706** | 0.787 | **0.871** | 0.816 | **0.790** | 0.320 | **0.363** | 3.219 | **3.495** |
| | RLINEAR | 0.799 | **0.961** | 0.682 | **0.750** | 0.731 | **0.839** | 0.799 | **0.815** | 0.341 | **0.366** | 3.104 | **3.349** |
| | PATCHTST | 0.941 | **1.070** | 0.805 | **0.873** | 0.785 | **0.861** | 0.840 | **0.855** | 0.360 | **0.395** | 3.104 | **3.519** |
| | iTRANSFORMER | 0.890 | **1.004** | 0.717 | **0.810** | 0.745 | **0.858** | 0.820 | **0.877** | 0.371 | **0.377** | 3.346 | **3.541** |
| | MAMBA | 0.917 | **1.141** | 0.855 | **0.908** | 0.792 | **1.006** | 0.834 | **0.933** | 0.384 | **0.405** | 3.452 | **3.854** |
| | BI-MAMBA+ | 0.975 | **1.081** | 0.813 | **0.876** | 0.850 | **0.928** | 0.926 | **0.943** | 0.393 | **0.426** | 3.744 | **3.967** |
| | MODERNTCN | 0.882 | **1.028** | 0.793 | **0.811** | 0.640 | **0.858** | 0.783 | **0.886** | 0.354 | **0.381** | 3.096 | **3.554** |
| | TCN | 0.835 | **1.014** | 0.790 | **0.829** | 0.794 | **0.907** | 0.791 | **0.872** | 0.372 | **0.383** | 3.352 | **3.525** |
| CSI 1000 | LSTM | 1.275 | **1.372** | 1.311 | **1.325** | 0.807 | **0.893** | 0.842 | **0.883** | 0.494 | **0.502** | 3.888 | **4.050** |
| | GRU | 1.321 | **1.410** | 1.283 | **1.397** | 0.862 | **0.889** | 0.836 | **0.971** | 0.498 | **0.525** | 4.192 | **4.272** |
| | DLINEAR | 1.107 | **1.204** | 1.021 | **1.223** | 0.635 | **0.796** | 0.620 | **0.938** | 0.446 | **0.477** | 3.450 | **4.067** |
| | RLINEAR | 1.091 | **1.207** | 1.067 | **1.183** | 0.730 | **0.827** | 0.827 | **0.921** | 0.445 | **0.480** | 3.423 | **3.872** |
| | PATCHTST | 1.290 | **1.389** | 1.310 | **1.349** | 0.796 | **0.871** | 0.851 | **0.926** | 0.469 | **0.502** | 3.685 | **4.016** |
| | iTRANSFORMER | 1.216 | **1.296** | 1.253 | **1.352** | 0.692 | **0.875** | 0.737 | **0.995** | 0.451 | **0.474** | 3.481 | **3.955** |
| | MAMBA | 1.328 | **1.376** | 1.292 | **1.331** | 0.866 | **0.873** | 0.906 | **0.907** | 0.492 | **0.510** | 3.966 | **4.134** |
| | BI-MAMBA+ | 1.337 | **1.377** | 1.227 | **1.284** | 0.723 | **0.927** | 0.709 | **0.968** | 0.468 | **0.510** | 3.622 | **4.409** |
| | MODERNTCN | 1.213 | **1.311** | 1.192 | **1.225** | 0.654 | **0.951** | 0.723 | **0.861** | 0.477 | **0.488** | 3.955 | **4.143** |
| | TCN | 1.072 | **1.219** | 0.953 | **1.127** | 0.837 | **0.902** | 0.903 | **0.909** | 0.491 | **0.511** | 4.148 | **4.407** |

## 5.1. Experimental Setup

**Data and Splitting.** We evaluate our method on large-scale real-world market data covering three major stock indices: CSI 300 (Large-cap), CSI 500 (Mid-cap), and CSI 1000 (Small-cap). This selection ensures the evaluation covers diverse market dynamics across different market capitalizations. The dataset spans from January 2019 to July 2025. To strictly prevent look-ahead bias, we employ a chronological split: Training (Jan 2019 – July 2023), Validation (July 2023 – July 2024), and Testing (July 2024 – July 2025). Additional datasets for evaluation on the US S&P 500 and historical stress-test periods are detailed in Appendix A.1.

**Implementation.** Input features are constructed from minute-level multivariate data. Models map the feature sequence to realized returns and are optimized using AdamW. During training, the supervision signal is dynamically determined by our bi-level optimization framework, which learns the optimal label horizon. Comprehensive details are provided in Appendix A.5.

## 5.2. Baselines and Metrics

**Baselines.** To ensure a comprehensive and rigorous evaluation, we benchmark our approach against a diverse set of state-of-the-art models. Specifically, we select ten representative baselines spanning five distinct deep forecasting families to cover a wide range of temporal modeling paradigms. These include **Linear-based models** (DLinear (Zeng et al., 2023) and RLinear (Li et al., 2023)), which utilize simple yet effective linear mappings; **RNN-based networks** (GRU (Dey & Salem, 2017) and LSTM (Hochreiter & Schmidhuber, 1997)), representing classical sequential modeling approaches; **CNN-based architectures** (TCN (Liu et al., 2019) and ModernTCN (Luo & Wang, 2024)), which capture local temporal dependencies through convolutions; **Transformer-based models** (PatchTST (Nie et al., 2022) and iTransformer (Liu et al., 2024b)), representing attention-based forecasting paradigms; and **SSM-based models** (Mamba (Gu & Dao, 2024) and Bi-Mamba+ (Liang et al., 2024)), representing the latest advancements in structured state space models.

**Metrics.** Performance is assessed across two dimensions: Predictive Signal Quality, measured by the Pearson correlation (IC) and Spearman rank correlation (RankIC), along with their stability ratios (ICIR, RankICIR); and Investment Potential, evaluated by the average daily return of the top 10% stocks, reporting the Daily Return and Sharpe Ratio.

Detailed descriptions of these baselines and metric calculations are provided in Appendix B.

## 5.3. Main Results

To align with standard benchmarks in the literature, we focus our primary evaluation in the main text on the Daily Close-to-Close setting (Scenario 1), comparing our framework against the conventional baseline trained strictly on the final target horizon $\mathbf{r}_t^\Delta$. As shown in Table 1, our bi-level method consistently outperforms the standard paradigm across all ten diverse architectures—ranging from classical RNNs and linear models to state-of-the-art Transformers and SSMs—and across all three market capitalization indices (CSI 300, 500, and 1000). Notably, our method not only enhances raw predictive accuracy (IC and RankIC) but also yields substantial improvements in signal stability (ICIR and RankICIR) and portfolio simulation metrics (Top Return and Sharpe Ratio). This universal improvement validates that relaxing the strict label-target alignment can fundamentally enhance representation learning in noisy financial environments.

Beyond standard interday predictions, we extend our evaluation to different temporal granularities. As detailed in Appendix E.1, the results under intraday scenarios perfectly align with our theoretical expectations. In the highly momentum-driven 30-minute window (Scenario 2), our method automatically recovers the canonical target, performing on par with the baseline. In the longer 90-minute window (Scenario 3), our adaptive horizon selection successfully captures the intermediate optimal signal, delivering notable gains in signal stability.

To ensure our findings are not artifacts of a specific market or historical period, we conduct extensive cross-market and out-of-distribution validations. Specifically, our framework maintains consistent performance improvements on the US S&P 500 index (Appendix E.3), confirming that the Label Horizon Paradox holds even in highly efficient markets. Furthermore, stress tests conducted during the severe market crash and continuous downtrend of 2024 (Appendix E.4) demonstrate that dynamic label selection provides crucial risk resilience under extreme macroeconomic shocks.

Finally, while statistical metrics are informative, real-world quantitative trading is constrained by market frictions. In Appendix E.5, we present a simple downstream portfolio backtest incorporating transaction costs, slippage, and TWAP execution. Our method consistently generates higher annualized returns while effectively reducing maximum drawdowns (MDD) across backbones. Crucially, as analyzed in Appendix F, these comprehensive performance and economic gains are achieved with only marginal computational overhead, thanks to the efficiency of our single-step inner-loop design.

*Table 2.* **Performance Comparison.** Experiments are conducted using an LSTM backbone across Scenarios 1, 2, and 3 on CSI 500. We compare our proposed method (*) against two baselines: Naive Averaging (†) and Equal-Weight Multi-Task Learning (‡).

| CONFIGURATION | IC($\times 10$) | ICIR | RANKIC($\times 10$) | RANKICIR |
|---|---|---|---|---|
| SCENARIO 1* | **1.029** | **0.861** | **0.859** | **0.895** |
| SCENARIO 1† | 0.969 | 0.778 | 0.821 | 0.817 |
| SCENARIO 1‡ | 0.932 | 0.803 | 0.814 | 0.837 |
| SCENARIO 2* | **1.491** | 1.396 | **1.943** | **2.062** |
| SCENARIO 2† | 1.453 | **1.471** | 1.857 | 2.021 |
| SCENARIO 2‡ | 1.435 | 1.456 | 1.849 | 1.998 |
| SCENARIO 3* | **1.082** | 1.095 | **1.399** | **1.609** |
| SCENARIO 3† | 1.050 | **1.125** | 1.359 | 1.538 |
| SCENARIO 3‡ | 1.071 | 1.118 | 1.367 | 1.596 |

# 6. Further Analysis

In this section, we conduct a series of in-depth analyses to dissect the internal mechanisms of our framework.

## 6.1. Necessity of Bi-level Optimization

A natural question arises: *can we achieve similar benefits by simply averaging potential labels or treating them as equal multi-task targets?* To investigate this, we compare our method against two baselines:

**1. Naive Averaging:** The model is trained on a single fixed label $\bar{\mathbf{y}}_t$, constructed as the arithmetic mean of all candidate proxy horizons ($\bar{\mathbf{y}}_t = \frac{1}{\Delta} \sum_\delta \mathbf{z}_t^\delta$).

**2. Equal-Weight MTL:** The model predicts all candidate horizons simultaneously using a multi-task learning objective with fixed, equal weights ($\mathcal{L} = \frac{1}{\Delta} \sum_\delta \ell(\hat{\mathbf{y}}_t, \mathbf{r}_t^\delta)$).

As shown in Table 2, our Bi-level approach still outperforms both methods. Naive Averaging tends to dilute the predictive signal, as it indiscriminately mixes high-quality intermediate horizons with noisy ones. Similarly, Equal-Weight MTL fails to prioritize, forcing the model to allocate capacity to horizons that may contain little learnable information. In contrast, our BLO framework dynamically up-weights the most effective horizons based on validation feedback, proving that adaptive selection is superior to blind aggregation. More detailed experiments and discussions are provided in the Appendix E.2.

## 6.2. Alignment of Learned Weights with the Optima

A central claim of our work is that the proposed method acts as an automatic signal-to-noise ratio detector. To verify this, we visualize the final distribution of the learned horizon weights $\boldsymbol{\lambda}$ and compare them against the empirical paradox curve (the actual test IC of models trained on fixed horizons, as observed in Section 3).

Figure 4 reveals a striking alignment. The peak of the

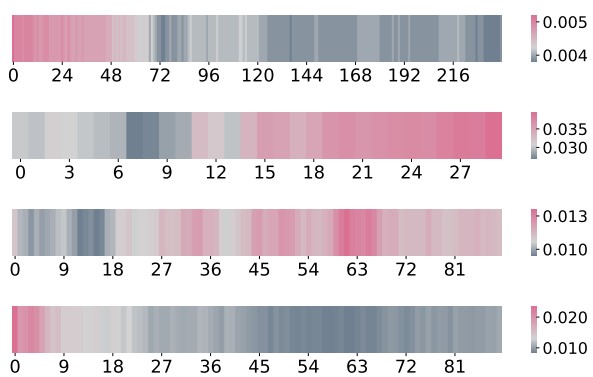

*Figure 4.* **Visualization of Learned Horizon Weights ($\lambda$).** We illustrate the final distribution of $\lambda$ learned by an LSTM model on the CSI500 dataset. The panels correspond to Scenario 1 (top), Scenario 2 (second), Scenario 3 (third), and Scenario 3 without warm-up (bottom).

learned weight distribution $\lambda$ coincides closely with the horizon that achieved the highest test IC in our brute-force grid search. This indicates that the outer-loop gradient successfully senses the signal-noise trade-off, navigating the optimization focus toward the sweet spot of supervision.

### 6.3. Impact of Warm-up Phase

As visualized in the bottom row of Figure 4, omitting the warm-up phase leads to a degenerate behavior in the label distribution: the learned weights $\lambda$ skew disproportionately toward the earliest horizons, effectively discarding the information embedded in longer timeframes.

From an optimization perspective, this phenomenon closely mirrors the concept of shortcut learning (Geirhos et al., 2020). Because short-horizon labels are temporally closer to the input features, they naturally exhibit stronger correlations. Consequently, they offer an "easy" path for rapid loss reduction. When the bi-level optimization begins from a completely untrained state, the single-step inner loop greedily exploits these highly correlated but ultimately myopic targets to minimize the immediate training objective.

However, while this greedy exploitation enables very fast initial convergence, it fundamentally limits the model's predictive capacity. By fixating solely on the easiest-to-predict short-term dynamics, the network fails to capture the broader, more robust market trends, thereby imposing a strictly low performance ceiling.

Therefore, the warm-up phase serves as a necessary structural prior. This ensures that when the dynamic weight optimization is finally engaged, the model evaluates the utility of different horizons based on meaningful feature representations, effectively preventing the bi-level optimization from getting trapped in trivial local minima.

## 7. Related Works

Our work addresses the inherent challenges of financial forecasting by rethinking the definition and optimization of supervision signals. Prior research primarily mitigates noise through robust feature extraction (Feng et al., 2019) or by applying noisy-label learning techniques (Zhang & Sabuncu, 2018; Reed et al., 2014) to construct cleaner training data (Zeng et al., 2024), yet these methods typically operate under a fixed prediction horizon. Consequently, they neglect the dynamic trade-off between signal realization and noise accumulation inherent in the label's temporal evolution.

It is also important to distinguish our approach from standard label smoothing techniques (Müller et al., 2019). While traditional label smoothing acts as a regularizer by softening the numerical distribution of a fixed target (e.g., converting hard one-hot labels to soft probabilities), our method operates orthogonally along the time axis. We do not alter the inherent cross-sectional distribution of the return labels; rather, we dynamically search for the optimal temporal horizon to serve as the supervision signal. Furthermore, while one might intuitively attempt to "smooth" labels by averaging targets across multiple horizons, our analysis in Section 6.1 demonstrates that our adaptive horizon selection is fundamentally distinct from, and strictly superior to, such naive temporal mixing.

To exploit this unexamined temporal dimension, we adopt a bi-level optimization framework. While such frameworks have been widely employed for sample re-weighting (Ren et al., 2018; Shu et al., 2019; Jiang et al., 2018) to filter noisy training data, we diverge by extending this paradigm from sample selection to label selection. Instead of cleaning input samples, our approach utilizes it to dynamically search for the optimal supervision horizon, effectively navigating the evolving signal-noise trade-off.

## 8. Conclusion

This study identifies the Label Horizon Paradox, challenging the conventional wisdom that training labels must strictly align with inference targets. We reveal that optimal generalization requires decoupling the two to balance signal emergence against market noise. By employing a bi-level framework to autonomously learn the optimal supervision horizon, our approach consistently enhances the performance of diverse existing architectures. Ultimately, this work establishes a new paradigm for dynamic, label-centric optimization in highly stochastic forecasting environments.

## Impact Statement

This research is intended solely for academic purposes. The methodologies, models, and empirical results presented herein do not constitute financial, legal, or investment advice, and the authors assume no responsibility for any financial losses or adverse consequences arising from their application in real-world trading environments. Furthermore, from an ethical data perspective, all experiments were conducted using strictly publicly available market data (e.g., CSI 300, 500, 1000, and the US S&P 500). No private, sensitive, or personally identifiable information (PII) of individual investors was utilized, ensuring full compliance with data privacy standards.

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

# A. Experimental Setup

## A.1. Data and Universe

Our empirical study is comprehensively evaluated across diverse market environments, including the Chinese A-share market and the US equity market. For our primary evaluation, we focus on three widely recognized A-share indices: CSI 300, CSI 500, and CSI 1000, which correspond to large-, medium-, and small-cap stocks, respectively, thereby ensuring that our dataset is highly representative of the broader market.

To ensure a robust evaluation and prevent look-ahead bias, we employ strict chronological splits. The data spans from January 2018 to July 2025, partitioned into three distinct settings based on the experimental scenarios:

- **Standard Setting (CSI 300/500/1000 & S&P 500):** Used for the main results and the US market generalization experiment.

    - **Training Set:** January 2019 – July 2023.
    - **Validation Set:** July 2023 – July 2024.
    - **Test Set:** July 2024 – July 2025.

- **Stress Test Setting (CSI 500):** Shifted to cover a severe and continuous macroeconomic downtrend, culminating in the February 2024 market crash.

    - **Training Set:** January 2018 – July 2022.
    - **Validation Set:** July 2022 – July 2023.
    - **Test Set:** July 2023 – July 2024.

## A.2. Raw Data

For each stock $i$ on day $t$, the raw data is an intraday multivariate time series consisting of 7 variables: Open Price, High Price, Low Price, Close Price, Amount, Volume, and Transaction Count. These variables are sampled at 1-minute intervals.

Depending on the market, the length of the daily sequence varies to match the respective trading hours:

- **A-share Market:** The sequence covers the 240 minutes of a standard trading day (9:30 AM to 11:30 AM and 1:00 PM to 3:00 PM, totaling 4 hours), denoted as $\mathbf{V}_{i,t} \in \mathbb{R}^{240 \times 7}$.

- **US Market:** The sequence is extended to cover the longer daily trading session (9:30 AM to 4:00 PM, totaling 6.5 hours), denoted as $\mathbf{V}_{i,t} \in \mathbb{R}^{390 \times 7}$.

## A.3. Feature Engineering

To maintain computational tractability while preserving microstructure information, we divide the raw sequence into non-overlapping patches. Each patch contains 15 minutes of trading data, from which we extract statistical descriptors to form the model input. Within each patch, we extract a diverse set of statistical descriptors to characterize the local market state from multiple dimensions. Specifically, for each of the raw variables, we compute a broad spectrum of univariate descriptors, including momentum and scale measures such as the arithmetic mean, the change rate, and the normalized range. To capture the distribution shape and potential fat-tail characteristics within the window, we further incorporate higher-order moments, exemplified by the standard deviation, skewness, and kurtosis. Beyond univariate analysis, we employ a wide variety of bivariate interaction descriptors to model the dynamic coupling between different market facets. These descriptors encompass linear coupling measures—for instance, the Pearson correlation coefficient for key pairs like price-volume—as well as relative intensity ratios, such as volume per transaction and other scale-invariant metrics. Due to the extensive nature of the feature set, we refrain from enumerating every specific indicator and focus here on these representative categories. All extracted features are concatenated into a consolidated tensor with $D$ dimensions, providing a high-fidelity representation of the microstructure for subsequent model input.

## A.4. Scenario-Specific Configuration

To strictly align with the three market scenarios defined in the main text, we construct specific input tensors based on the decision time $t$. Let $D$ denote the feature dimension per patch. The input configurations are defined as follows (Scenario 1 is evaluated on both the A-share and the US S&P 500 markets, whereas Scenarios 2 and 3 are considered only for the A-share market):

**Scenario 1: Interday (Standard) Prediction.** The decision time $t$ corresponds to the market close. The input $\mathbf{x}_{i,t}$ aggregates the entire trading day's microstructure information. For the A-share market, this consists of 16 patches (covering the full 240-minute trading session), resulting in an input tensor shape of $\mathbf{x}_{i,t} \in \mathbb{R}^{16 \times D}$. For the US S&P 500 market, this consists of 26 patches (covering the 390-minute session), resulting in $\mathbf{x}_{i,t} \in \mathbb{R}^{26 \times D}$. The objective is to predict the return of the next trading day ($\Delta = 1$ day), calculated based on the last price of the continuous session to preserve temporal continuity.

**Scenario 2: Intraday (30-minute) Prediction.** The decision time $t$ is set at the midpoint of the trading session (11:30, Morning Close), leveraging the market's distinct morning/afternoon session structure. To prevent look-ahead bias, the input utilizes only the morning session data (09:30–11:30), resulting in a sequence of 8 patches. The input tensor shape is $\mathbf{x}_{i,t} \in \mathbb{R}^{8 \times D}$. The objective is to predict the return over the subsequent 30-minute interval ($\Delta = 30$ min).

**Scenario 3: Intraday (90-minute) Prediction.** The decision time $t$ is also set at the midpoint of the trading session (11:30). Similar to Scenario 2, the input is derived exclusively from the morning session, maintaining an identical input tensor shape of $\mathbf{x}_{i,t} \in \mathbb{R}^{8 \times D}$. However, the forecasting objective here is to predict the return over the subsequent 90-minute interval ($\Delta = 90$ min), capturing a longer intraday trend than Scenario 2.

## A.5. Implementation Details

### A.5.1. PREDICTIVE MODEL TRAINING

Input features are constructed from minute-level multivariate market data. The predictive model maps this feature sequence to the designated label (i.e., the return at the specific horizon determined by the experimental setting).

**Training Configuration.** We set the batch to cover 20 trading days, i.e., one batch contains data from a 20-day window. All models are optimized using the AdamW optimizer with MSE loss. Importantly, we standardize both the model outputs and the labels cross-sectionally (zero mean and unit variance). Under this normalization, minimizing the MSE is equivalent to maximizing the Pearson correlation (IC) between predictions and labels, since for standardized variables

$$\mathrm{MSE}(\hat{y}, y) = \mathbb{E}[(\hat{y} - y)^2] = 2 - 2\,\mathrm{Corr}(\hat{y}, y), \tag{15}$$

so both objectives are aligned up to an affine transformation.

**Early Stopping Strategy.** We employ an early stopping mechanism to prevent overfitting. During training, the loss on the validation set is monitored at the end of every epoch. The training process is terminated if the validation loss does not improve for 5 consecutive epochs (patience = 5). Upon termination, the model parameters corresponding to the lowest validation loss are restored as the final model.

### A.5.2. BI-LEVEL OPTIMIZATION TRAINING.

The training of our Label-Horizon-based bi-level framework proceeds in two stages: a warm-up stage and a bi-level iteration stage.

**Warm-up Stage.** Before initiating the alternating optimization of the model parameters and the horizon parameter, we perform a warm-up phase to stabilize the model weights. The warm-up lasts for $N_{warm} = 3$ epochs. During this phase, the training configuration (learning rate, batch construction) is identical to the standard predictive model training described above.

**Bi-level Iteration Stage.** Following the warm-up, we proceed with the bi-level updates. To strictly separate the data used for the inner loop (model update) and the outer loop (horizon update), we employ a date-based random splitting strategy:

- **Data Splitting:** For each batch containing 20 trading days, the days are randomly divided into two equal subsets (10 days each). The first subset serves as the support set (for the inner loop), and the second subset serves as the query set

for the outer loop.

- **Inner Loop (Model Update):** The model parameters are updated using the support set. The weights $\lambda$ are obtained by normalizing a set of learnable parameters via a softmax transformation. Since the model has been pre-trained during the warmup phase, we use a reduced learning rate $1 \times 10^{-6}$ for fine-tuning.

- **Outer Loop (Horizon Update):** The horizon parameter is updated using the query set. The learning rate for the outer loop is set to $1 \times 10^{-3}$ and the weight of the entropy term is set to $1 \times 10^{-3}$.

# B. Baselines and Metrics

## B.1. Baseline Models

To ensure the robustness of our findings and position our method within the broader landscape of deep time-series forecasting, we benchmark across five distinct architectural families. These baselines range from classical sequence models to the latest state-of-the-art foundation models:

- **RNN-based Methods:** We include GRU and LSTM. These recurrent architectures serve as the traditional workhorses for financial sequence modeling, processing data sequentially to capture temporal dependencies, though often struggling with long-term memory retention.

- **Linear/MLP-based Methods:** Despite the rise of complex architectures, simple linear models have shown surprising effectiveness in noisy time-series tasks. We compare against DLinear, which employs a trend-seasonal decomposition combined with linear layers, and RLinear, which focuses on reversible normalization to handle distribution shifts.

- **CNN-based Methods:** To evaluate convolutional approaches, we select TCN, which utilizes dilated causal convolutions to model long-range history with a receptive field that grows exponentially. We also include ModernTCN, a recent adaptation that incorporates large-kernel convolutions and parameter-efficient designs.

- **Transformer-based Methods:** we employ PatchTST and iTransformer. PatchTST segments time series into patches and applies channel-independent attention to capture local semantic patterns, while iTransformer inverts the attention mechanism to model the correlation between multivariate variates directly, making it particularly relevant for capturing cross-stock interactions.

- **SSM-based Methods:** We assess the emerging class of Selective State Space Models (SSMs) via Mamba and Bi-Mamba+. These models utilize a hardware-efficient selection mechanism to achieve linear computational complexity relative to sequence length, theoretically allowing for superior modeling of long contexts without the quadratic cost of Transformers.

## B.2. Evaluation Metrics

We evaluate model performance using a comprehensive set of metrics that assess both the statistical predictive power and the practical economic value of the generated signals.

**Predictive Accuracy Metrics.** These metrics measure the correlation between the model's output signal and the ground-truth future returns, focusing on the signal's information content.

- **IC (Information Coefficient):** Defined as the Pearson correlation coefficient between the predicted scores and the realized returns across the cross-section of stocks at each time step. We report the time-series mean of the daily IC.

- **ICIR (Information Ratio):** A measure of prediction stability, calculated as the ratio of the mean IC to the standard deviation of the IC (Mean(IC)/Std(IC)). A higher ICIR indicates a more consistent signal performance.

- **RankIC & RankICIR:** Given that financial returns often contain outliers, we also report the Spearman rank correlation (RankIC) and its corresponding stability ratio (RankICIR). Rank-based metrics are robust to extreme values and more accurately reflect the sorting capability required for portfolio construction.

**Portfolio Simulation Metrics.** To gauge the model's predictive power, we focus on the performance of the top-ranked stocks. Specifically, at each decision time, we calculate the equal-weighted average daily return of the top 10% of stocks with the highest predicted scores.

- **Top Returns:** The average daily return of the top 10% of stocks ranked by predicted values, where, for all three forecasting scenarios, we generate predictions once per day and compute this metric using the corresponding horizon-specific realized returns.

- **Sharpe Ratio:** The risk-adjusted return, calculated as the mean of the daily returns divided by the standard deviation of these daily returns. This metric indicates the daily return generated per unit of risk. In our experiments, we report the annualized Sharpe Ratio by multiplying the daily Sharpe by $\sqrt{252}$.

While our primary evaluation focuses on the predictive quality and stability of the learned signals to cleanly elucidate the Label Horizon Paradox, we also recognize the importance of practical economic value. Therefore, in addition to the Top Returns and Sharpe Ratio metrics evaluated directly on the signals, we conduct a simple downstream portfolio backtesting simulation—incorporating transaction costs, slippage, and TWAP execution—detailed in Appendix E.5.

# C. Theoretical Analysis with a Linear Factor Model

In this section, we provide a rigorous theoretical justification for the Label Horizon Paradox. We construct a theoretical framework grounded in the Arbitrage Pricing Theory (APT). We begin with the standard static APT model and extend it into a continuous-time setting to explicitly capture the dynamics of signal realization and noise accumulation.

Our goal is to derive the generalization performance (Information Coefficient) of a linear estimator trained on a proxy horizon return $r^\delta$ and evaluated on the final target horizon return $r^\Delta$. For notational brevity, we omit the decision time subscript $t$ and the stock index $i$ in the subsequent analysis.

## C.1. Data Generating Process: A Time-Varying APT Extension

We consider a universe of stocks where returns are driven by observable factors.

### C.1.1. THE STANDARD STATIC APT

In the classical APT framework, the return of a stock over a fixed period is decomposed into a systematic component driven by common factors $\mathbf{s}$ and an idiosyncratic component. Using our notation:

$$r^\Delta = \mathbf{w}^{*\top}\mathbf{s} + \epsilon^\Delta, \tag{16}$$

where $\mathbf{s}$ represents factor exposures, $\mathbf{w}^*$ represents factor risk premia, and $\epsilon^\Delta$ is the idiosyncratic noise. This model is typically static—it describes the return over a single, undefined period where information is assumed to be fully reflected.

### C.1.2. TEMPORAL EXTENSION

We extend the standard APT by introducing a continuous time parameter $\delta \in (0, \Delta]$ to model the trajectory of returns.

First, we formally define the signal and weight components with the following assumptions:

- **Factor Exposure $\mathbf{s}$:** Let $\mathbf{s} \in \mathbb{R}^d$ be the vector of factor exposures (predictive signals) for a stock at the decision time $t$. We assume factors are whitened such that $\mathbb{E}[\mathbf{s}\mathbf{s}^\top] = \mathbf{I}_d$.

- **True Factor Loadings $\mathbf{w}^*$:** Let $\mathbf{w}^* \in \mathbb{R}^d$ represent the latent, ground-truth linear relationship between factors and returns. We assume $\|\mathbf{w}^*\|_2 = 1$ for identifiability (a normalization that fixes the scale between $\alpha(\delta)$ and $\mathbf{w}^*$ and simplifies the derivations).

For any cumulative return $r^\delta$ from the decision time $t$ to $t + \delta$, we propose the following time-varying specification:

$$r^\delta = \underbrace{\alpha(\delta)}_{\text{Signal Realization}} \mathbf{w}^{*\top}\mathbf{s} + \underbrace{\epsilon^\delta}_{\text{Accumulated Noise}}. \tag{17}$$

This formulation can be interpreted as a snapshot of the APT model at a specific horizon $\delta$. Compared to the standard baseline, we introduce two critical time-dependent modifications:

1. **Signal Realization Process ($\alpha(\delta)$):** Unlike the standard APT, which assumes equilibrium (i.e., information is fully priced), we acknowledge that information incorporation takes time. Let $\alpha : (0, \Delta] \to (0, A]$ be a monotonically increasing function representing the degree of price discovery, where $A > 0$ is a finite upper bound determined by the overall scale of the latent signal.

   - $\alpha(\delta) < A$ implies partial underreaction or gradual diffusion of information.
   - $\alpha(\delta) \approx A$ implies the signal is (effectively) fully realized at that horizon.

2. **Noise Accumulation Process ($\epsilon^\delta$):** We explicitly model the idiosyncratic term $\epsilon^\delta$ as a dynamic process rather than a static error. Consistent with the Random Walk Hypothesis (Brownian motion) for efficient markets, we assume the noise is jointly defined across horizons as a microstructure-shifted Brownian motion:

$$\epsilon^\delta = \eta_0 + \sigma W_\delta, \qquad \eta_0 \sim \mathcal{N}(0, \sigma^2 \delta_0), \;\; W_\delta \text{ is a standard Brownian motion}, \tag{18}$$

where $\sigma^2$ is the accumulation rate of idiosyncratic volatility, and $\delta_0 > 0$ represents intrinsic microstructure noise that exists even as $\delta \to 0$. This implies the marginal distribution

$$\epsilon^\delta \sim \mathcal{N}\big(0, \sigma^2(\delta + \delta_0)\big), \tag{19}$$

and for any $0 < \delta \leq \Delta$, the nested random-walk property

$$\mathrm{Cov}(\epsilon^\delta, \epsilon^\Delta) = \mathrm{Var}(\epsilon^\delta) = \sigma^2(\delta + \delta_0). \tag{20}$$

While this assumption is generally reasonable at short horizons such as intraday (minute-level) and daily intervals, it may become less accurate over much longer horizons where structural breaks and slow-moving factors dominate. Since our study focuses on short-term stock forecasting at minute and daily frequencies, the random-walk-based noise model is well aligned with the forecasting regimes of interest.

**Relationship to Standard APT:** For any fixed horizon $\delta$, our model collapses to a standard linear factor model with effective signal strength $\alpha(\delta)\mathbf{w}^*$ and noise variance $\sigma^2(\delta + \delta_0)$. Structurally, this formulation assumes a *horizon-invariant* signal direction: the latent predictive relationship $\mathbf{w}^*$ remains constant, while the temporal dynamics are entirely captured by the scalar realization function $\alpha(\delta)$. While real-world factor exposures might exhibit rotations across different forecasting horizons, this deliberate simplification serves a crucial theoretical purpose. By fixing the signal direction, we mathematically isolate the fundamental temporal trade-off without the confounding effects of factor shifting. Consequently, the core of the Label Horizon Paradox naturally emerges purely from the dynamic interplay between the derivative of $\alpha(\delta)$ (signal accumulation) and the derivative of the noise variance (noise accumulation). Extending this framework to accommodate more complex dynamics, such as horizon-dependent factor rotations, remains a valuable direction for future research.

### C.2. The Learning Setup: Finite-Sample OLS

Consider a training dataset $\mathcal{D} = \{(\mathbf{s}_j, r_j^\delta)\}_{j=1}^N$ of size $N$, where the labels are realized returns at the proxy horizon $\delta$. We employ Ordinary Least Squares (OLS) to estimate the factor loadings.

The estimated weight vector $\hat{\mathbf{w}}_\delta$ is given by:

$$\hat{\mathbf{w}}_\delta = (\mathbf{S}^\top \mathbf{S})^{-1} \mathbf{S}^\top \mathbf{r}^\delta = \alpha(\delta)\mathbf{w}^* + (\mathbf{S}^\top \mathbf{S})^{-1} \mathbf{S}^\top \epsilon^\delta. \tag{21}$$

Assuming $N$ is sufficiently large such that $\mathbf{S}^\top \mathbf{S} \approx N \mathbf{I}_d$ (due to the whitening assumption), the estimator decomposes into:

$$\hat{\mathbf{w}}_\delta = \alpha(\delta)\mathbf{w}^* + \mathbf{z}_\delta, \quad \text{where } \mathbf{z}_\delta \sim \mathcal{N}\left(\mathbf{0}, \frac{\sigma^2(\delta + \delta_0)}{N}\mathbf{I}_d\right). \tag{22}$$

The variance of the estimation error $\mathbf{z}_\delta$ grows linearly with $\delta$, reflecting the difficulty of learning from long-horizon labels dominated by accumulated random-walk noise.

### C.3. Derivation of Generalization Performance (Final IC)

We define the Information Coefficient (IC) as the Pearson correlation between the model's prediction $\hat{y} = \hat{\mathbf{w}}_\delta^\top \mathbf{s}$ and the final target return $r^\Delta$, evaluated on an independent test set.

First, we derive the necessary variance and covariance terms:

- **Covariance:** Recall that

$$\hat{y}^\delta = \hat{\mathbf{w}}_\delta^\top \mathbf{s} = \big(\alpha(\delta)\mathbf{w}^* + \mathbf{z}_\delta\big)^\top \mathbf{s} = \alpha(\delta)\mathbf{w}^{*\top}\mathbf{s} + \mathbf{z}_\delta^\top \mathbf{s},$$

and the target can be written as

$$r^\Delta = \alpha(\Delta)\mathbf{w}^{*\top}\mathbf{s} + \epsilon^\Delta.$$

By construction, the estimation noise $\mathbf{z}_\delta$ is independent of the test-time features $\mathbf{s}$ and the test noise $\epsilon^\Delta$, and has zero mean. Using these facts and the whitening assumption $\mathbb{E}[\mathbf{s}\mathbf{s}^\top] = \mathbf{I}_d$, together with $\|\mathbf{w}^*\|_2 = 1$, we have

$$\mathrm{Cov}(\hat{y}^\delta, r^\Delta) = \mathrm{Cov}\big(\alpha(\delta)\mathbf{w}^{*\top}\mathbf{s} + \mathbf{z}_\delta^\top \mathbf{s}, \ \alpha(\Delta)\mathbf{w}^{*\top}\mathbf{s} + \epsilon^\Delta\big) \tag{23}$$

$$= \alpha(\delta)\alpha(\Delta)\,\mathrm{Var}(\mathbf{w}^{*\top}\mathbf{s}) + \underbrace{\mathrm{Cov}(\mathbf{z}_\delta^\top \mathbf{s}, \mathbf{w}^{*\top}\mathbf{s})}_{=0} + \underbrace{\mathrm{Cov}(\alpha(\delta)\mathbf{w}^{*\top}\mathbf{s}, \epsilon^\Delta)}_{=0} + \underbrace{\mathrm{Cov}(\mathbf{z}_\delta^\top \mathbf{s}, \epsilon^\Delta)}_{=0} \tag{24}$$

$$= \alpha(\delta)\alpha(\Delta) \cdot 1 = \alpha(\delta)\alpha(\Delta). \tag{25}$$

Therefore, only the systematic signal component contributes to the covariance:

$$\text{Cov}(\hat{y}^\delta, r^\Delta) = \text{Cov}(\alpha(\delta)\mathbf{w}^{*\top}\mathbf{s}, \alpha(\Delta)\mathbf{w}^{*\top}\mathbf{s}) = \alpha(\delta)\alpha(\Delta). \tag{26}$$

- **Prediction Variance ($V_{\text{est}}$):**
$$V_{\text{est}}(\delta) = \text{Var}(\hat{\mathbf{w}}_\delta^\top \mathbf{s}) = \alpha(\delta)^2 + \frac{d}{N}\sigma^2(\delta + \delta_0). \tag{27}$$

- **Target Variance ($V_{\text{target}}$):**
$$V_{\text{target}} = \text{Var}(r^\Delta) = \alpha(\Delta)^2 + \sigma^2(\Delta + \delta_0). \tag{28}$$

Combining these, the expected squared IC on the final target is:

$$J(\delta) \triangleq \text{IC}_{\text{final}}^2(\delta) = \frac{\text{Cov}(\hat{y}^\delta, r^\Delta)^2}{V_{\text{est}}(\delta)V_{\text{target}}} = \frac{\alpha(\delta)^2\alpha(\Delta)^2}{[\alpha(\delta)^2 + K(\delta + \delta_0)] \cdot V_{\text{target}}}, \tag{29}$$

where $K = \frac{d}{N}\sigma^2$ is a constant.

## C.4. Proof of the Paradox

To determine the optimal training horizon $\delta^*$, we analyze the behavior of the expected squared IC, $J(\delta)$. Since the target variance $V_{\text{target}}$ and $\alpha(\Delta)$ are constant scalars independent of the training horizon $\delta$, maximizing $J(\delta)$ is equivalent to maximizing the log-objective of $\alpha(\delta)^2 / [\alpha(\delta)^2 + K(\delta + \delta_0)]$.

### C.4.1. INTUITIVE PROOF: SIGNAL VS. NOISE ACCUMULATION

We analyze the log-performance, which allows for an additive decomposition of the trade-off. Ignoring constant terms, the objective function is:

$$\ln J(\delta) = \underbrace{2\ln\alpha(\delta)}_{\text{Information Gain}} - \underbrace{\ln[\alpha(\delta)^2 + K(\delta + \delta_0)]}_{\text{Noise Penalty}} + C. \tag{30}$$

This decomposition reveals the structural mechanism behind the Label Horizon Paradox:

1. **Information Gain:** This term represents the logarithmic accumulation of the realized signal. It increases monotonically as $\delta$ grows.

2. **Noise Penalty:** This term represents the penalty from the total prediction variance. Since the idiosyncratic noise $K(\delta + \delta_0)$ follows a random walk, this term grows strictly and indefinitely.

**The Paradox Mechanism:** At short horizons, the rapid realization of the signal dominates the noise, leading to performance gains. However, as the horizon extends, signal growth naturally decelerates (diminishing returns), while noise accumulation remains constant and linear. Therefore, a tipping point $\delta^*$ is eventually reached where the steady accumulation of noise overwhelms the benefit of the slowing signal, causing the final performance to decline.

### C.4.2. MATHEMATICAL DERIVATIVE ANALYSIS

To determine the location of the optimal horizon $\delta^*$, we examine the first derivative of the log-performance function $J(\delta)$. Differentiating Eq. (30) with respect to $\delta$ yields the gradient:

$$\frac{d}{d\delta}\ln J(\delta) = \frac{2\alpha'(\delta)}{\alpha(\delta)} - \frac{2\alpha(\delta)\alpha'(\delta) + K}{\alpha(\delta)^2 + K(\delta + \delta_0)}. \tag{31}$$

To understand the sign of this derivative, we analyze the condition for performance improvement, i.e., $\frac{d}{d\delta}\ln J(\delta) > 0$. Substituting Eq. (31) into this inequality gives:

$$\frac{2\alpha'(\delta)}{\alpha(\delta)} > \frac{2\alpha(\delta)\alpha'(\delta) + K}{\alpha(\delta)^2 + K(\delta + \delta_0)}. \tag{32}$$

Assuming $\alpha(\delta) > 0$ and $K > 0$, we can cross-multiply by the denominators without changing the inequality sign:

$$2\alpha'(\delta)\left[\alpha(\delta)^2 + K(\delta + \delta_0)\right] > \alpha(\delta)\left[2\alpha(\delta)\alpha'(\delta) + K\right]. \tag{33}$$

Expanding both sides reveals a common term:

$$2\alpha'(\delta)\alpha(\delta)^2 + 2\alpha'(\delta)K(\delta + \delta_0) > 2\alpha(\delta)^2\alpha'(\delta) + \alpha(\delta)K. \tag{34}$$

Subtracting the common term $2\alpha'(\delta)\alpha(\delta)^2$ from both sides, the inequality simplifies significantly:

$$2\alpha'(\delta)K(\delta + \delta_0) > \alpha(\delta)K. \tag{35}$$

Finally, dividing by $K$ and rearranging the terms to separate the signal dynamics from the time horizon, we obtain the necessary and sufficient condition for the derivative to be positive:

$$\frac{\alpha'(\delta)}{\alpha(\delta)} > \frac{1}{2(\delta + \delta_0)}. \tag{36}$$

This inequality compares the relative growth rate of the signal ($\frac{\alpha'}{\alpha}$) with the hyperbolic decay rate of the noise horizon ($\frac{1}{2(\delta+\delta_0)}$).

### C.5. Detailed Mechanism Analysis of Different Horizon Scenarios

In this section, we apply the rigorous derivative analysis derived above to interpret the empirical results presented in Section 3.1. As established in Eq. (36), the shape of the performance curve $J(\delta)$ is determined entirely by the competition between the relative signal growth rate and the inverse time horizon. The sign of the gradient depends on the condition:

$$\underbrace{\frac{\alpha'(\delta)}{\alpha(\delta)}}_{\text{Signal Growth Rate}} \gtrless \underbrace{\frac{1}{2(\delta + \delta_0)}}_{\text{Noise Threshold}}. \tag{37}$$

**Scenario 1: Interday Prediction (Monotonic Decrease).** In the standard daily prediction setting (predicting close-to-close returns), empirical results consistently favor the shortest proxy (close-to-open). Theoretically, this implies that the relevant information is priced in almost immediately at the market open.

- **Mathematical Regime:** Since the signal saturates early, $\alpha(\delta) \approx$ const and $\alpha'(\delta) \approx 0$ for almost all $\delta > 0$.

- **Inequality Analysis:** The Signal Growth Rate vanishes ($\approx 0$) while the Noise Threshold remains positive. Thus, $\frac{\alpha'}{\alpha} < \frac{1}{2(\delta+\delta_0)}$ holds for the entire duration.

- **Conclusion:** The derivative is strictly negative, rendering the shortest horizon optimal ($\delta^* \to 0$).

**Scenario 2: Intraday 30-minute Prediction (Monotonic Increase).** For short-term 30-minute windows, the market undergoes active price discovery driven by momentum that persists throughout the interval.

- **Mathematical Regime:** The signal $\alpha(\delta)$ grows robustly across the short window, maintaining a high marginal gain $\alpha'(\delta)$.

- **Inequality Analysis:** The short duration keeps the Noise Threshold ($\frac{1}{2(\delta+\delta_0)}$) comparable to the signal growth. Because the interval is too short for the signal to saturate, the inequality $\frac{\alpha'}{\alpha} > \frac{1}{2(\delta+\delta_0)}$ remains true up to $\delta = \Delta$.

- **Conclusion:** The derivative remains positive, suggesting that the model benefits from extending the proxy horizon to the full target length ($\delta^* = \Delta$).

**Scenario 3: Intraday 90-minute Prediction (Hump-Shaped).** When the target window extends to 90 minutes, we observe the characteristic Label Horizon Paradox. This scenario represents the transition between active information diffusion and signal saturation.

- **Mathematical Regime:** Initially, information flows rapidly ($\alpha'$ is large). However, as $\delta$ increases, the predictive validity of the signal at time $t$ naturally decays or is fully incorporated into the price ($\alpha' \to 0$).

- **Inequality Analysis:**

  1. Early Phase: The rapid signal uptake ensures $\frac{\alpha'}{\alpha} > \frac{1}{2(\delta+\delta_0)}$, driving performance up.

  2. Late Phase: As signal growth slows, $\frac{\alpha'}{\alpha}$ drops below the threshold $\frac{1}{2(\delta+\delta_0)}$, meaning the marginal noise cost exceeds the marginal information value.

- **Conclusion:** The gradient crosses from positive to negative at an intermediate point. The optimal horizon $\delta^*$ is precisely the instant where the relative signal growth equals the inverse time horizon.

## C.6. Decomposition under Interday Prediction Scenario

We now apply the theoretical framework to Scenario 1 (Standard Daily Close-to-Close Prediction). In this setting, the input features are historical data available at the close of day $t$. Given the overnight information processing period, the predictive signal derived from these features is typically incorporated into prices immediately at the market open of day $t + 1$.

Mathematically, this corresponds to the regime of rapid price discovery. The signal realization function satisfies $\alpha(\delta) \approx$ const and $\alpha'(\delta) \approx 0$ for any horizon $\delta$ extending beyond the market open, and in particular $\alpha(\delta) \approx \alpha(\Delta)$ across the relevant horizons.

Under the general model $r^\delta = \alpha(\delta)\mathbf{w}^{*\top}\mathbf{s} + \epsilon^\delta$ and $r^\Delta = \alpha(\Delta)\mathbf{w}^{*\top}\mathbf{s} + \epsilon^\Delta$, and recalling that the idiosyncratic noise follows a nested random-walk structure such that $\mathrm{Cov}(\epsilon^\delta, \epsilon^\Delta) = \mathrm{Var}(\epsilon^\delta)$, the covariance between the proxy label $r^\delta$ and the target $r^\Delta$ is

$$\mathrm{Cov}(r^\delta, r^\Delta) = \alpha(\delta)\alpha(\Delta)\,\mathrm{Var}(\mathbf{w}^{*\top}\mathbf{s}) + \mathrm{Cov}(\epsilon^\delta, \epsilon^\Delta) = \alpha(\delta)\alpha(\Delta) + \sigma^2(\delta + \delta_0). \tag{38}$$

The proxy and target variances are given by

$$V_{\mathrm{proxy}}(\delta) = \mathrm{Var}(r^\delta) = \alpha(\delta)^2 + \sigma^2(\delta + \delta_0), \tag{39}$$

$$V_{\mathrm{target}} = \mathrm{Var}(r^\Delta) = \alpha(\Delta)^2 + \sigma^2(\Delta + \delta_0). \tag{40}$$

The corresponding correlations are

$$\rho(\hat{y}^\delta, r^\delta) = \frac{\mathrm{Cov}(\hat{y}^\delta, r^\delta)}{\sqrt{V_{\mathrm{est}}}\sqrt{V_{\mathrm{proxy}}}} = \frac{\alpha(\delta)^2}{\sqrt{V_{\mathrm{est}}}\sqrt{V_{\mathrm{proxy}}}}, \tag{41}$$

$$\rho(r^\delta, r^\Delta) = \frac{\mathrm{Cov}(r^\delta, r^\Delta)}{\sqrt{V_{\mathrm{proxy}}}\sqrt{V_{\mathrm{target}}}} = \frac{\alpha(\delta)\alpha(\Delta) + \sigma^2(\delta + \delta_0)}{\sqrt{V_{\mathrm{proxy}}}\sqrt{V_{\mathrm{target}}}}, \tag{42}$$

and

$$\mathrm{IC}_{\mathrm{final}}(\delta) = \rho(\hat{y}^\delta, r^\Delta) = \frac{\mathrm{Cov}(\hat{y}^\delta, r^\Delta)}{\sqrt{V_{\mathrm{est}}}\sqrt{V_{\mathrm{target}}}} = \frac{\alpha(\delta)\alpha(\Delta)}{\sqrt{V_{\mathrm{est}}}\sqrt{V_{\mathrm{target}}}}. \tag{43}$$

Combining these, the final Information Coefficient admits the following general multiplicative form:

$$\mathrm{IC}_{\mathrm{final}}(\delta) = \rho(\hat{y}^\delta, r^\delta)\,\rho(r^\delta, r^\Delta)\,\frac{\alpha(\delta)\alpha(\Delta)}{\alpha(\delta)^2} \cdot \frac{V_{\mathrm{proxy}}(\delta)}{\alpha(\delta)\alpha(\Delta) + \sigma^2(\delta + \delta_0)}. \tag{44}$$

In the interday rapid price discovery regime, we have $\alpha(\delta) \approx \alpha(\Delta)$ for all relevant proxy horizons. Substituting this into the general expressions, the covariance between the proxy label and the target simplifies to

$$\mathrm{Cov}(r^\delta, r^\Delta) = \alpha(\delta)\alpha(\Delta) + \sigma^2(\delta + \delta_0) \approx \alpha(\delta)^2 + \sigma^2(\delta + \delta_0) = V_{\mathrm{proxy}}(\delta). \tag{45}$$

Under this approximation, the proxy effectively behaves as a noisy but nearly unbiased version of the target at the same signal realization level, and the final IC is numerically close to the product of the model's fit to the proxy and the proxy's correlation with the target:

$$\text{IC}_{\text{final}}(\delta) \approx \rho(\hat{y}^{\delta}, r^{\delta}) \times \rho(r^{\delta}, r^{\Delta}). \tag{46}$$

where:

- $\rho(\hat{y}^{\delta}, r^{\delta})$: Represents the **Proxy IC** (how well the model learns the specific label $r^{\delta}$).

- $\rho(r^{\delta}, r^{\Delta})$: Represents the **Label Alignment** (how well the proxy $r^{\delta}$ correlates with the ultimate target $r^{\Delta}$).

To rigorously validate this theoretical decomposition, we conducted an extensive empirical study. We trained independent LSTM models across the full spectrum of intraday horizons at minute-level granularity. To ensure statistical robustness and mitigate initialization noise, each horizon-specific model was trained using multiple random seeds. This evaluation was performed across three major indices (CSI300, CSI500, and CSI1000).

For each model, we computed the actual test IC ($\text{IC}_{\text{final}}$) and compared it against the product of the empirically measured components $\rho(\hat{y}^{\delta}, r^{\delta})$ and $\rho(r^{\delta}, r^{\Delta})$. As illustrated in Figure 3, the empirical results demonstrate a near-perfect alignment between the theoretical decomposition and the actual performance. This validates that in the interday regime, the performance dynamics are indeed governed by the fundamental trade-off between signal saturation and random walk noise accumulation, as predicted by our modified APT framework.

## D. Alternative Derivation via Partial Correlation Formula

In the main text, we derived the performance decomposition identity using a structural Linear Factor Model. In this appendix, we provide an alternative derivation based purely on the statistical properties of correlation. Specifically, we show that the decomposition can be rigorously understood as a special case of the Partial Correlation Formula where the residual term vanishes due to the specific signal dynamics of the market. This shows that the multiplicative decomposition of the final IC is not an artifact of the specific factor-model setup, but a direct consequence of the vanishing partial correlation between the model prediction and the residual return beyond the proxy horizon. For notational brevity, we omit the decision time subscript $t$ and the stock index $i$ in the subsequent analysis.

### D.1. The Decomposition and the Vanishing Residual

From a statistical perspective, the relationship between the model prediction $\hat{y}^\delta$, the proxy label $r^\delta$, and the final target $r^\Delta$ can, under the standard linear/Gaussian framework, be expressed via the following correlation decomposition:

$$\rho_{\hat{y}^\delta, r^\Delta} = \underbrace{\rho_{\hat{y}^\delta, r^\delta} \cdot \rho_{r^\delta, r^\Delta}}_{\text{Mediated Path}} + \underbrace{\rho_{\hat{y}^\delta, r^\Delta \cdot r^\delta} \sqrt{1 - \rho_{\hat{y}^\delta, r^\delta}^2} \sqrt{1 - \rho_{r^\delta, r^\Delta}^2}}_{\text{Residual Term}}. \tag{47}$$

Here, the notation is rigorously defined as follows:

- $\rho_{A,B}$ denotes the standard Pearson correlation coefficient between variables $A$ and $B$, consistent with the definition of IC used throughout the paper.

- $\rho_{A,B \cdot C}$ denotes the partial correlation coefficient between $A$ and $B$ given a control variable $C$. As formally derived in the subsequent section, this is defined as the Pearson correlation between the residuals of $A$ and $B$ after the linear effect of $C$ has been regressed out (i.e., $\rho(e_{A|C}, e_{B|C})$).

The physical interpretation of this formula in our context provides deep insight into the Label Horizon Paradox:

- The Mediated Path represents the efficacy of the prediction insofar as it captures information already contained in the proxy horizon $\delta$.

- The Residual Term is controlled by the partial correlation $\rho_{\hat{y}^\delta, r^\Delta \cdot r^\delta}$. This term measures the correlation between the model's prediction and the final target after the influence of the proxy $r^\delta$ has been removed. Effectively, it asks: *Can the model predict the return evolution from $\delta$ to $\Delta$ that is orthogonal to the return up to $\delta$?*

**Application to Scenario 1 (Interday Prediction):** In Scenario 1, the input features are derived from history prior to the market open. Due to the high efficiency of the market, the predictive signal contained in these historical features is typically priced in almost immediately upon the open.

Consequently, the subsequent price movement from the intermediate horizon $\delta$ to the final target $\Delta$ is dominated by new, idiosyncratic information and noise that was not available at the decision time. Since this future noise is strictly unforecastable based on the input features, the model's predictive power for this residual component is negligible.

Mathematically, this implies $\rho_{\hat{y}^\delta, r^\Delta \cdot r^\delta} \approx 0$. Substituting this into Eq. (47), the entire residual term vanishes, and we recover the multiplicative decomposition presented in the main text:

$$\rho_{\hat{y}^\delta, r^\Delta} \approx \rho_{\hat{y}^\delta, r^\delta} \cdot \rho_{r^\delta, r^\Delta}. \tag{48}$$

### D.2. Proof of the Partial Correlation Formula

For completeness, we provide the step-by-step algebraic proof of the general formula used above.

**1. Definition via Residuals.** The partial correlation $\rho_{X,Y \cdot Z}$ between two random variables $X$ and $Y$ given a controlling variable $Z$ is defined as the Pearson correlation between their residuals after linearly regressing out $Z$.

Without loss of generality, assume that $X$, $Y$, and $Z$ are standardized variables with zero mean and unit variance (i.e., $\mathbb{E}[X] = 0, \mathbb{E}[X^2] = 1$). The linear regression of $X$ on $Z$ and $Y$ on $Z$ can be expressed as:

$$\hat{X} = \rho_{X,Z}Z, \quad e_{X|Z} = X - \hat{X} = X - \rho_{X,Z}Z \tag{49}$$

$$\hat{Y} = \rho_{Y,Z}Z, \quad e_{Y|Z} = Y - \hat{Y} = Y - \rho_{Y,Z}Z \tag{50}$$

where $\rho_{X,Z}$ and $\rho_{Y,Z}$ correspond to the regression coefficients (slopes) in the standardized case.

**2. Deriving the Standard Formula.** The partial correlation is the correlation of the residuals $e_{X|Z}$ and $e_{Y|Z}$:

$$\rho_{X,Y\cdot Z} = \frac{\mathbb{E}[e_{X|Z}e_{Y|Z}]}{\sqrt{\mathbb{E}[e^2_{X|Z}]}\sqrt{\mathbb{E}[e^2_{Y|Z}]}}. \tag{51}$$

**Step 2.1: The Numerator (Covariance of Residuals).**

$$\mathbb{E}[e_{X|Z}e_{Y|Z}] = \mathbb{E}[(X - \rho_{X,Z}Z)(Y - \rho_{Y,Z}Z)] \tag{52}$$

$$= \mathbb{E}[XY - X\rho_{Y,Z}Z - Y\rho_{X,Z}Z + \rho_{X,Z}\rho_{Y,Z}Z^2] \tag{53}$$

$$= \underbrace{\mathbb{E}[XY]}_{\rho_{X,Y}} - \rho_{Y,Z}\underbrace{\mathbb{E}[XZ]}_{\rho_{X,Z}} - \rho_{X,Z}\underbrace{\mathbb{E}[YZ]}_{\rho_{Y,Z}} + \rho_{X,Z}\rho_{Y,Z}\underbrace{\mathbb{E}[Z^2]}_{1} \tag{54}$$

$$= \rho_{X,Y} - \rho_{X,Z}\rho_{Y,Z} - \rho_{X,Z}\rho_{Y,Z} + \rho_{X,Z}\rho_{Y,Z} \tag{55}$$

$$= \rho_{X,Y} - \rho_{X,Z}\rho_{Y,Z}. \tag{56}$$

**Step 2.2: The Denominator (Variance of Residuals).** Since the residual variance is $1 - R^2$:

$$\mathbb{E}[e^2_{X|Z}] = 1 - \rho^2_{X,Z}, \quad \mathbb{E}[e^2_{Y|Z}] = 1 - \rho^2_{Y,Z}. \tag{57}$$

Combining these, we recover the standard recursive formula:

$$\rho_{X,Y\cdot Z} = \frac{\rho_{X,Y} - \rho_{X,Z}\rho_{Y,Z}}{\sqrt{1 - \rho^2_{X,Z}}\sqrt{1 - \rho^2_{Y,Z}}}. \tag{58}$$

**3. Rearranging for the Decomposition Formula.** We solve Eq. (58) for the total correlation $\rho_{X,Y}$:

$$\rho_{X,Y\cdot Z}\sqrt{1 - \rho^2_{X,Z}}\sqrt{1 - \rho^2_{Y,Z}} = \rho_{X,Y} - \rho_{X,Z}\rho_{Y,Z} \tag{59}$$

$$\rho_{X,Y} = \rho_{X,Z}\rho_{Y,Z} + \rho_{X,Y\cdot Z}\sqrt{1 - \rho^2_{X,Z}}\sqrt{1 - \rho^2_{Y,Z}}. \tag{60}$$

**4. Application to the Label Horizon Problem.** Finally, we substitute the variables from our main context ($X \leftarrow \hat{y}^\delta$, $Y \leftarrow r^\Delta$, $Z \leftarrow r^\delta$) to obtain the formula in Eq. (47).

# E. Supplementary Results

## E.1. Results of Intraday Scenario

In the main text, we primarily focused on the standard interday prediction task (Scenario 1), which corresponds to the widely used close-to-close setting in academic studies. To further assess the generality of our framework under different temporal granularities, we report here the detailed experimental results for the two intraday scenarios defined in Section 3.1:

- **Scenario 2:** Intraday prediction with a 30-minute horizon ($\Delta = 30$ minutes).

- **Scenario 3:** Intraday prediction with a 90-minute horizon ($\Delta = 90$ minutes).

The experimental setup (data, feature construction, train/validation/test splits), the set of ten backbone architectures, and the six evaluation metrics are kept identical to those used in Scenario 1 to ensure a fair comparison.

Tables 3 and 4 summarize the results for Scenario 2 and Scenario 3, respectively.

### E.1.1. SCENARIO 2 (30-MINUTE HORIZON)

From Table 3, we observe that our bi-level framework and the standard training baseline achieve very similar performance across all three indices and all ten backbones. On most metrics, the two methods are within a narrow margin of each other, and the winner alternates depending on the specific model–dataset combination.

This outcome is fully consistent with the empirical pattern observed in Figure 2. In Scenario 2, the performance curve as a function of the training horizon is monotonically increasing, and the empirically optimal horizon $\delta^*$ is essentially aligned with the final target horizon $\Delta$. Our bi-level procedure therefore learns to concentrate its weight near the target horizon, effectively recovering the canonical choice. As a consequence, adaptively selecting the supervision horizon offers little additional benefit over directly training on the final target, and the two approaches perform on par, with model-specific fluctuations.

### E.1.2. SCENARIO 3 (90-MINUTE HORIZON)

For the longer intraday horizon in Scenario 3, Table 4 reveals a different picture. On many configurations, our method achieves higher IC, RankIC, and Top Return than the standard baseline, indicating that adaptively shifting the supervision away from the final horizon can indeed enhance the raw predictive signal. At the same time, there are a few exceptions where the baseline slightly outperforms our method on these point-estimate metrics.

This mixed pattern is again aligned with the empirical phenomenon in Figure 2. In Scenario 3, the performance curve is hump-shaped: the optimal horizon $\delta^*$ lies somewhere between 0 and $\Delta$, but the advantage of this intermediate horizon over the final horizon is modest, and becomes visible mainly after Gaussian smoothing of the noisy empirical curve. In a realistic high-noise financial environment, such a weak edge can be partially obscured by stochastic variability, so it is natural to see some configurations where training directly on the final target remains competitive.

However, a more consistent advantage of our method emerges when we examine the stability metrics: ICIR, RankICIR, and Sharpe Ratio. Across all three indices and most backbones in Scenario 3, our bi-level framework yields higher ICIR and RankICIR than the standard baseline, and also improves the Sharpe Ratio in a largely uniform manner. This suggests that even when the average IC gain is modest, adaptively selecting the supervision horizon helps the model produce signals that are more stable over time and more robust to noise, which is crucial for practical portfolio construction.

## E.2. Extended Analysis on the Necessity of Bi-level Optimization

In the main text, Section 6.1, we evaluate the necessity of bi-level optimization by comparing our method (training with the selected best single label) against two label aggregation baselines: Naive Averaging (†) and Equal-Weight Multi-Task Learning (‡). While our approach achieves the best overall performance, we observe that in Scenario 2 and Scenario 3, the ICIR obtained from a single selected label is slightly lower than that of models trained on aggregated labels.

However, this comparison is inherently conservative with respect to our method, because training on a single horizon label is naturally more volatile and statistically less stable than training on aggregated or multi-task targets that effectively average out noise across multiple horizons. To provide a more fair comparison, we further leverage the information contained in the

*Table 3.* **Results of Scenario 2.** Comprehensive performance comparison between standard training (Std.) and our Bi-level framework (Ours) across three market indices. All results are averaged over 5 random seeds. Bold indicates the better performance.

| DATASET | MODEL | IC (×10) | | ICIR | | RANKIC (×10) | | RANKICIR | | TOP RET (%) | | SHARPE RATIO | |
|---|---|---|---|---|---|---|---|---|---|---|---|---|---|
| | | STD. | OURS | STD. | OURS | STD. | OURS | STD. | OURS | STD. | OURS | STD. | OURS |
| CSI 300 | LSTM | 1.223 | **1.243** | **0.745** | 0.716 | 1.558 | **1.604** | **1.210** | 1.184 | 0.074 | **0.074** | 3.512 | **3.715** |
| | GRU | 1.195 | **1.225** | **0.723** | 0.690 | 1.666 | **1.693** | **1.144** | 1.107 | 0.078 | **0.080** | 3.900 | **4.106** |
| | DLINEAR | **1.208** | 1.183 | 0.689 | **0.722** | **1.612** | 1.592 | 1.197 | **1.258** | 0.072 | **0.075** | **3.951** | 3.799 |
| | RLINEAR | **1.194** | 1.193 | **0.819** | 0.786 | 1.519 | **1.536** | **1.218** | 1.158 | 0.071 | **0.074** | 3.547 | **3.739** |
| | PATCHTST | 1.176 | **1.281** | 0.835 | **0.855** | 1.434 | **1.630** | 1.202 | **1.311** | 0.070 | **0.078** | 3.832 | **3.995** |
| | ITRANSFORMER | **1.279** | 1.239 | 0.811 | **0.855** | **1.641** | 1.571 | **1.346** | 1.323 | 0.076 | **0.078** | 3.910 | **3.936** |
| | MAMBA | **1.268** | 1.259 | 0.823 | **0.844** | 1.616 | **1.658** | 1.297 | **1.328** | 0.079 | **0.084** | 3.812 | **4.552** |
| | BI-MAMBA+ | **1.242** | 1.220 | 0.851 | **0.855** | **1.591** | 1.536 | **1.303** | 1.299 | **0.082** | 0.072 | **4.332** | 3.623 |
| | MODERNTCN | **1.251** | 1.234 | **0.716** | 0.707 | **1.583** | 1.521 | **1.256** | 1.235 | **0.076** | 0.072 | **3.995** | 3.950 |
| | TCN | 1.229 | **1.240** | **0.729** | 0.691 | 1.597 | **1.605** | **1.254** | 1.238 | 0.079 | **0.084** | 4.110 | **4.123** |
| CSI 500 | LSTM | 1.474 | **1.491** | 1.372 | **1.396** | 1.922 | **1.943** | 1.931 | **2.062** | 0.121 | **0.122** | **5.172** | 5.076 |
| | GRU | 1.508 | **1.515** | 1.334 | **1.347** | **1.978** | 1.946 | 2.041 | **2.094** | **0.127** | 0.126 | 5.290 | **5.325** |
| | DLINEAR | **1.514** | 1.504 | **1.417** | 1.394 | 1.915 | **1.933** | **2.111** | 1.925 | **0.127** | 0.121 | **5.350** | 5.122 |
| | RLINEAR | 1.468 | **1.496** | **1.381** | 1.345 | 1.881 | **1.932** | 2.024 | **2.096** | **0.127** | 0.126 | **5.278** | 5.163 |
| | PATCHTST | **1.498** | 1.470 | **1.471** | 1.308 | **1.956** | 1.925 | **2.180** | 1.970 | 0.127 | **0.128** | 5.301 | **5.391** |
| | ITRANSFORMER | 1.472 | **1.480** | 1.246 | **1.372** | **1.950** | 1.947 | 1.940 | **2.032** | 0.118 | **0.120** | **5.188** | 5.051 |
| | MAMBA | **1.514** | 1.512 | **1.394** | 1.346 | 1.947 | **1.956** | **2.034** | 2.007 | **0.126** | 0.124 | **5.317** | 5.260 |
| | BI-MAMBA+ | **1.526** | 1.477 | 1.305 | **1.372** | **2.014** | 1.892 | 1.973 | **2.064** | **0.131** | 0.123 | **5.646** | 5.005 |
| | MODERNTCN | **1.481** | 1.472 | 1.339 | **1.359** | 1.905 | **1.933** | 1.930 | **1.978** | 0.124 | **0.126** | 5.173 | **5.228** |
| | TCN | **1.533** | 1.516 | **1.394** | 1.367 | **1.954** | 1.920 | **2.102** | 2.021 | 0.128 | **0.131** | 5.310 | **5.373** |
| CSI 1000 | LSTM | **1.161** | 1.156 | **1.100** | 1.066 | **1.910** | 1.852 | **1.889** | 1.873 | **0.115** | 0.114 | **4.418** | 3.908 |
| | GRU | **1.166** | 1.160 | **1.141** | 1.052 | 1.839 | **1.898** | **1.983** | 1.935 | 0.108 | **0.114** | 4.042 | **4.481** |
| | DLINEAR | 1.167 | **1.189** | **1.138** | 1.112 | 1.804 | **1.888** | 1.828 | **1.840** | 0.108 | **0.114** | 4.070 | **4.317** |
| | RLINEAR | **1.181** | 1.165 | **1.170** | 1.066 | 1.863 | **1.933** | **1.941** | 1.881 | 0.109 | **0.114** | 4.140 | **4.464** |
| | PATCHTST | 1.144 | **1.146** | 1.107 | **1.113** | 1.781 | **1.873** | 1.755 | **1.912** | 0.109 | **0.115** | 3.813 | **4.496** |
| | ITRANSFORMER | **1.203** | 1.170 | 1.111 | **1.143** | **1.914** | 1.827 | **1.865** | 1.856 | 0.113 | **0.114** | **4.465** | 4.251 |
| | MAMBA | 1.182 | **1.188** | 1.039 | **1.085** | 1.961 | 1.945 | 1.809 | **1.932** | 0.110 | **0.115** | 4.301 | **4.427** |
| | BI-MAMBA+ | 1.187 | 1.187 | 1.073 | **1.145** | **1.958** | 1.855 | 1.822 | **1.978** | **0.114** | 0.110 | **4.488** | 4.098 |
| | MODERNTCN | 1.147 | **1.178** | **1.127** | 1.098 | 1.694 | **1.844** | 1.851 | **1.956** | 0.107 | **0.112** | 3.960 | **4.187** |
| | TCN | 1.210 | **1.225** | 1.059 | **1.232** | **1.947** | 1.825 | 1.831 | **1.998** | **0.115** | 0.111 | **4.468** | 4.153 |

learned horizon weights $\lambda$:

- We first identify the top-5 horizons with the largest weights in $\lambda$.

- Using only these top-5 selected horizons, we then construct:
  1. A Naive Averaging variant (* †): train a model on the arithmetic mean of these top-5 labels.
  2. An Equal-Weight MTL variant (* ‡): train a model using only these top-5 horizons with equal weights in the multi-task loss.

In other words, we retain our bi-level optimization to select informative horizons, but then train baselines that aggregate or jointly model only these selected labels, instead of all candidates. This design removes the unfair advantage of aggregating over many noisy horizons, while still allowing label smoothing through averaging or multi-task learning.

Table 5 reports the performance of these additional configurations, again using an LSTM backbone on CSI 500 under Scenarios 1, 2, and 3. The results show that, once we restrict baselines to the top-5 horizons identified by our bi-level procedure, the resulting models consistently outperform their counterparts trained on all horizons.

### E.3. Generalization to US Markets (S&P 500)

To confirm that our proposed framework represents a general supervision principle rather than an artifact of a specific market, we extend our evaluation to the US equity market. Specifically, we conduct new experiments on the S&P 500 index. This evaluation strictly follows the Scenario 1 (daily prediction) protocol from the main paper, including data structure, feature engineering, and label construction. The dataset is chronologically split into January 2019–July 2023 (Train), July 2023–July 2024 (Validation), and July 2024–July 2025 (Test). The only adjustment made is a longer input sequence to accommodate the longer daily trading session characteristic of the US market.

*Table 4.* **Results of Scenario 3.** Comprehensive performance comparison between standard training (Std.) and our Bi-level framework (Ours) across three market indices. All results are averaged over 5 random seeds. Bold indicates the better performance.

| Dataset | Model | IC (×10) | | ICIR | | RankIC (×10) | | RankICIR | | Top Ret (%) | | Sharpe Ratio | |
|---|---|---|---|---|---|---|---|---|---|---|---|---|---|
| | | Std. | Ours | Std. | Ours | Std. | Ours | Std. | Ours | Std. | Ours | Std. | Ours |
| CSI 300 | LSTM | 0.937 | **0.947** | 0.530 | **0.666** | **1.245** | 1.136 | 0.886 | **1.010** | 0.039 | **0.046** | 1.114 | **1.240** |
| | GRU | 0.928 | **0.934** | 0.560 | **0.578** | 1.228 | **1.241** | 0.891 | **0.936** | 0.029 | **0.049** | 0.843 | **1.421** |
| | DLinear | **0.865** | 0.834 | 0.511 | **0.669** | **1.144** | 1.087 | 0.933 | **1.002** | 0.027 | **0.055** | 0.760 | **1.635** |
| | RLinear | 0.877 | **0.904** | 0.632 | **0.655** | 1.117 | **1.191** | 0.959 | **1.020** | 0.038 | **0.041** | 1.016 | **1.137** |
| | PatchTST | 0.867 | **0.976** | 0.636 | **0.671** | 1.002 | **1.216** | 0.852 | **1.007** | 0.033 | **0.044** | 0.953 | **1.287** |
| | iTransformer | 0.891 | **0.950** | 0.593 | **0.639** | 1.162 | **1.175** | 1.003 | **1.046** | 0.042 | **0.061** | 1.167 | **1.785** |
| | Mamba | 0.949 | **1.043** | 0.634 | **0.669** | 1.144 | **1.308** | 0.967 | **1.081** | 0.043 | **0.059** | 1.216 | **1.769** |
| | Bi-Mamba+ | 0.966 | **1.058** | 0.605 | **0.655** | 1.225 | **1.283** | 0.970 | **0.980** | 0.047 | **0.061** | 1.317 | **1.823** |
| | ModernTCN | 0.950 | **1.059** | 0.517 | **0.543** | 1.210 | **1.407** | 0.809 | **0.862** | 0.033 | **0.055** | 1.031 | **1.790** |
| | TCN | 0.856 | **1.002** | **0.625** | 0.556 | 1.002 | **1.317** | **0.966** | 0.933 | 0.031 | **0.053** | 0.883 | **1.630** |
| CSI 500 | LSTM | 1.069 | **1.082** | 1.019 | **1.095** | **1.407** | 1.399 | 1.563 | **1.609** | **0.083** | 0.080 | **2.096** | 1.809 |
| | GRU | 1.052 | **1.080** | 1.016 | **1.221** | 1.414 | **1.435** | 1.499 | **1.694** | 0.074 | **0.081** | 1.661 | **1.923** |
| | DLinear | **1.038** | 0.991 | 1.022 | **1.064** | **1.277** | 1.241 | **1.525** | 1.458 | 0.072 | **0.078** | 1.658 | **1.889** |
| | RLinear | 0.935 | **1.023** | 1.099 | **1.228** | 1.153 | **1.283** | 1.459 | **1.590** | 0.068 | **0.079** | 1.494 | **1.772** |
| | PatchTST | **1.047** | 1.036 | 0.975 | **1.232** | **1.397** | 1.306 | 1.442 | **1.657** | 0.086 | **0.091** | 1.975 | **1.999** |
| | iTransformer | **1.072** | 1.039 | 1.119 | **1.233** | **1.387** | 1.344 | 1.639 | **1.733** | 0.085 | **0.092** | 2.044 | **2.069** |
| | Mamba | 1.083 | **1.108** | 1.056 | **1.209** | 1.397 | **1.405** | 1.554 | **1.670** | 0.081 | **0.086** | 1.877 | **1.952** |
| | Bi-Mamba+ | 1.075 | **1.100** | 1.053 | **1.222** | 1.331 | **1.431** | 1.548 | **1.646** | 0.078 | **0.089** | 1.725 | **2.103** |
| | ModernTCN | 0.996 | **1.027** | 0.986 | **1.024** | 1.176 | **1.407** | 1.537 | **1.659** | 0.074 | **0.080** | 1.918 | **2.069** |
| | TCN | 1.081 | **1.105** | 1.047 | **1.178** | 1.328 | **1.381** | 1.516 | **1.657** | 0.078 | **0.092** | 1.765 | **2.141** |
| CSI 1000 | LSTM | 0.884 | **0.928** | 0.850 | **1.012** | 1.244 | **1.447** | 1.550 | **1.553** | 0.076 | **0.076** | 1.514 | **1.631** |
| | GRU | 0.903 | **0.907** | 0.888 | **1.069** | **1.389** | 1.333 | 1.605 | **1.651** | 0.068 | **0.080** | 1.404 | **1.675** |
| | DLinear | **0.924** | 0.891 | 0.873 | **1.136** | **1.385** | 1.310 | 1.477 | **1.635** | **0.075** | 0.067 | **1.647** | 1.609 |
| | RLinear | 0.861 | **0.906** | 0.975 | **0.998** | 1.225 | **1.321** | 1.494 | **1.574** | 0.062 | **0.074** | 1.257 | **1.537** |
| | PatchTST | 0.899 | **0.924** | **0.995** | 0.967 | 1.236 | **1.366** | 1.590 | **1.599** | 0.065 | **0.083** | 1.310 | **1.737** |
| | iTransformer | **0.948** | 0.942 | 0.929 | **1.029** | **1.318** | 1.311 | 1.550 | **1.620** | 0.074 | **0.077** | 1.557 | **1.627** |
| | Mamba | 0.894 | **0.908** | 0.909 | **1.058** | 1.309 | **1.333** | 1.577 | **1.765** | 0.070 | **0.078** | 1.469 | **1.610** |
| | Bi-Mamba+ | **0.955** | 0.931 | 0.898 | **0.962** | **1.440** | 1.346 | **1.657** | 1.596 | **0.085** | 0.083 | **1.877** | 1.719 |
| | ModernTCN | 0.916 | **0.941** | 0.872 | **0.979** | 1.311 | **1.374** | 1.515 | **1.529** | 0.074 | **0.078** | 1.503 | **1.718** |
| | TCN | 0.932 | **0.951** | 0.960 | **1.048** | 1.304 | **1.327** | 1.569 | **1.630** | 0.074 | **0.083** | 1.539 | **1.711** |

The US market is widely recognized as highly efficient, meaning that information is absorbed into prices much faster than in emerging markets. According to our theoretical framework (Section 3.2), a faster signal realization rate ($\alpha'(\delta) \to 0$ earlier) implies a more rapid dominance of the noise penalty, leading to a faster performance decay over the prediction horizon. As shown in Table 6, our bi-level framework consistently outperforms the standard paradigm across all ten backbone architectures. This confirms that the Label Horizon Paradox exists even in highly efficient markets, and our adaptive horizon learning effectively navigates the rapid signal-noise trade-off inherent in the US market.

### E.4. Stress Tests During Market Crashes

To further evaluate the risk resilience and temporal generalizability of our framework, we conducted a stress test focusing on a period of extreme market volatility. Specifically, we evaluate our models on the CSI 500 index with a shifted timeframe to capture a period of severe and continuous macroeconomic downtrend, culminating in the well-known market crash in February 2024.

Similar to the US market evaluation, this stress test strictly follows the Scenario 1 protocol regarding data structure, feature engineering, and label construction. The dataset is chronologically split into January 2018–July 2022 (Train), July 2022–July 2023 (Validation), and July 2023–July 2024 (Test, which covers the aforementioned market crash).

As shown in Table 7, despite the extreme market stress and distribution shifts, models trained with our bi-level optimization framework consistently maintain their performance improvements over the standard baselines. This robust performance demonstrates that dynamic label selection not only enhances signal quality in normal market conditions but also provides crucial stability when the market undergoes severe structural shocks.

*Table 5.* **Impact of Horizon Selection on Aggregation and MTL Performance.** Experiments are conducted using an LSTM backbone across Scenarios 1, 2, and 3 on CSI 500. We compare: (i) training with all candidate horizons (Naive Averaging: †; Equal-Weight MTL: ‡), and (ii) training with only the top-5 horizons selected by the learned horizon weights $\lambda$ (Naive Averaging on top-5: *†; Equal-Weight MTL on top-5: *‡). The results show that using the bi-level-selected top-5 horizons consistently improves over using all horizons.

| CONFIGURATION | IC($\times 10$) | ICIR | RANKIC($\times 10$) | RANKICIR | TOP RET (%) | SHARPE RATIO |
|---|---|---|---|---|---|---|
| SCENARIO 1* | 1.029 | 0.861 | 0.859 | 0.895 | 0.383 | 3.660 |
| SCENARIO 1† | 0.969 | 0.778 | 0.821 | 0.817 | 0.374 | 3.516 |
| SCENARIO 1*† | **1.066** | **0.865** | **0.902** | **0.921** | **0.398** | **3.682** |
| SCENARIO 1‡ | 0.932 | 0.803 | 0.814 | 0.837 | 0.380 | 3.377 |
| SCENARIO 1*‡ | 0.973 | 0.821 | 0.860 | 0.842 | 0.377 | 3.420 |
| SCENARIO 2* | **1.491** | 1.396 | **1.943** | 2.062 | 0.122 | 5.076 |
| SCENARIO 2† | 1.453 | 1.471 | 1.857 | 2.021 | 0.121 | 4.933 |
| SCENARIO 2*† | 1.486 | **1.613** | 1.892 | **2.104** | **0.126** | 5.089 |
| SCENARIO 2‡ | 1.435 | 1.456 | 1.849 | 1.998 | 0.123 | 4.931 |
| SCENARIO 2*‡ | 1.483 | 1.479 | 1.935 | 2.097 | 0.124 | **5.095** |
| SCENARIO 3* | 1.082 | 1.095 | 1.399 | 1.609 | 0.080 | 1.809 |
| SCENARIO 3† | 1.050 | 1.125 | 1.359 | 1.538 | 0.082 | 1.935 |
| SCENARIO 3*† | **1.086** | **1.247** | **1.464** | 1.642 | **0.087** | **2.073** |
| SCENARIO 3‡ | 1.071 | 1.118 | 1.367 | 1.596 | 0.085 | 1.965 |
| SCENARIO 3*‡ | 1.083 | 1.181 | 1.448 | **1.669** | 0.085 | 1.975 |

*Table 6.* **Results on the US Market (S&P 500).** Comprehensive performance comparison under the daily prediction setting. All results are averaged over 5 random seeds. Bold indicates the better performance.

| DATASET | MODEL | IC ($\times 10$) | | ICIR | | RANKIC ($\times 10$) | | RANKICIR | | TOP RET (%) | | SHARPE RATIO | |
|---|---|---|---|---|---|---|---|---|---|---|---|---|---|
| | | STD. | OURS | STD. | OURS | STD. | OURS | STD. | OURS | STD. | OURS | STD. | OURS |
| **S&P 500** | LSTM | 0.314 | **0.459** | 0.302 | **0.441** | 0.149 | **0.219** | 0.139 | **0.206** | 0.141 | **0.159** | 1.443 | **1.600** |
| | GRU | 0.346 | **0.399** | 0.333 | **0.351** | 0.181 | **0.202** | 0.181 | **0.184** | **0.164** | 0.157 | 1.560 | **1.618** |
| | DLINEAR | 0.326 | **0.506** | 0.332 | **0.472** | 0.139 | **0.213** | 0.159 | **0.215** | 0.125 | **0.166** | 1.368 | **1.733** |
| | RLINEAR | 0.352 | **0.461** | 0.335 | **0.452** | 0.170 | **0.199** | 0.171 | **0.214** | 0.155 | **0.159** | 1.544 | **1.614** |
| | PATCHTST | 0.309 | **0.489** | 0.347 | **0.421** | **0.180** | 0.178 | 0.169 | **0.216** | 0.137 | **0.148** | 1.330 | **1.570** |
| | ITRANSFORMER | 0.397 | **0.499** | 0.402 | **0.435** | 0.215 | **0.228** | 0.187 | **0.232** | **0.177** | 0.175 | 1.712 | **2.097** |
| | MAMBA | 0.429 | **0.527** | 0.406 | **0.539** | 0.192 | **0.241** | 0.180 | **0.245** | 0.160 | **0.175** | 1.640 | **1.875** |
| | BI-MAMBA+ | 0.315 | **0.473** | 0.236 | **0.417** | 0.177 | **0.225** | 0.124 | **0.182** | 0.149 | **0.164** | 1.409 | **1.653** |
| | MODERNTCN | 0.306 | **0.455** | 0.247 | **0.383** | 0.141 | **0.198** | 0.111 | **0.171** | 0.136 | **0.155** | 1.231 | **1.486** |
| | TCN | 0.423 | **0.540** | 0.388 | **0.515** | 0.200 | **0.250** | 0.210 | **0.227** | 0.164 | **0.168** | 1.677 | **1.686** |

## E.5. Downstream Portfolio Backtesting

To further demonstrate the practical application value of our framework, we extend our evaluation beyond statistical predictive metrics by conducting a simple downstream portfolio backtesting experiment. This simulation incorporates realistic trading constraints to assess the actual economic value generated by the predictive signals.

**Experimental Setup.** Following standard quantitative investment paradigms, we construct a daily rebalanced, equal-weight portfolio. The detailed settings are as follows:

- **Stock Universe & Period:** The backtest is conducted on the constituents of the CSI 500 index over the test period from July 2024 to July 2025.

- **Execution Strategy:** To prevent look-ahead bias and simulate realistic execution, the models generate predictions 15 minutes before the market close. We then use the Time-Weighted Average Price (TWAP) of the subsequent 10 minutes as the execution price for all trades.

- **Portfolio Construction:** On each trading day, we select the top 20% of stocks with the highest predicted scores and assign them equal weights in the portfolio.

*Table 7.* **Stress Test Results on CSI 500 (Jul 2023 – Jul 2024).** Performance comparison during a severe market downtrend and crash. All results are averaged over 5 random seeds. Bold indicates the better performance.

| DATASET | MODEL | IC (×10) | | ICIR | | RANKIC (×10) | | RANKICIR | | TOP RET (%) | | SHARPE RATIO | |
|---|---|---|---|---|---|---|---|---|---|---|---|---|---|
| | | STD. | OURS | STD. | OURS | STD. | OURS | STD. | OURS | STD. | OURS | STD. | OURS |
| CSI 500 | LSTM | 0.874 | **0.909** | **0.927** | 0.916 | 0.817 | **0.932** | 1.013 | **1.104** | 0.129 | **0.131** | 1.411 | **1.454** |
| | GRU | 0.883 | **0.890** | 0.887 | **0.928** | 0.843 | **0.926** | 1.016 | **1.053** | 0.123 | **0.129** | **1.414** | 1.341 |
| | DLINEAR | 0.746 | **0.817** | 0.801 | **0.817** | 0.749 | **0.865** | 0.914 | **1.015** | 0.103 | **0.125** | 1.164 | **1.362** |
| | RLINEAR | 0.879 | **0.884** | 0.921 | **0.979** | 0.897 | **0.924** | **1.117** | 1.086 | **0.130** | 0.126 | 1.367 | **1.458** |
| | PATCHTST | 0.884 | **0.931** | 0.857 | **0.861** | 0.862 | **0.974** | 0.928 | **1.005** | 0.128 | **0.136** | 1.387 | **1.469** |
| | iTRANSFORMER | 0.846 | **0.910** | 0.828 | **0.868** | 0.871 | **0.946** | 0.985 | **1.032** | 0.106 | **0.129** | 1.176 | **1.438** |
| | MAMBA | 0.922 | **0.965** | 0.927 | **0.982** | 0.915 | **1.011** | 1.055 | **1.083** | 0.126 | **0.139** | 1.402 | **1.511** |
| | BI-MAMBA+ | 0.953 | **0.957** | **0.930** | 0.906 | 0.896 | **0.990** | 0.977 | **1.069** | 0.138 | **0.141** | 1.495 | **1.542** |
| | MODERNTCN | 0.740 | **0.847** | 0.744 | **0.823** | 0.698 | **0.868** | 0.762 | **0.929** | 0.089 | **0.119** | 0.981 | **1.319** |
| | TCN | 0.935 | **0.951** | 0.940 | **0.978** | 0.929 | **0.964** | 1.049 | **1.097** | 0.129 | **0.134** | 1.397 | **1.451** |

- **Real-world Constraints:** Transaction costs (commissions and stamp duties) and execution slippage are strictly incorporated into the backtesting framework to accurately reflect real-world trading frictions.

As shown in Table 8, applying our bi-level optimization method consistently improves downstream trading performance across almost all backbone architectures. Notably, our framework not only enhances the Annualized Return (Ann. Ret) but also effectively reduces the Maximum Drawdown (MDD) and Annualized Volatility (Ann. Vol) in most cases. Consequently, the risk-adjusted returns, measured by the Sharpe Ratio, exhibit substantial and consistent improvements over the standard training paradigm.

*Table 8.* **Downstream Portfolio Backtesting Results on CSI 500.** Performance comparison incorporating real-world constraints such as transaction costs and TWAP execution. Bold indicates the better performance.

| DATASET | MODEL | ANN. RET (%) | | ANN. VOL (%) | | SHARPE RATIO | | MDD (%) | |
|---|---|---|---|---|---|---|---|---|---|
| | | STD. | OURS | STD. | OURS | STD. | OURS | STD. | OURS |
| CSI 500 | LSTM | 42.18 | **42.27** | 25.89 | **24.53** | 1.63 | **1.72** | -13.96 | **-11.25** |
| | GRU | 36.15 | **40.90** | 25.01 | **24.68** | 1.45 | **1.66** | -13.55 | **-12.17** |
| | DLINEAR | 34.41 | **41.65** | 25.11 | **24.80** | 1.37 | **1.68** | -12.84 | **-12.01** |
| | RLINEAR | 33.97 | **41.04** | **23.87** | 24.95 | 1.42 | **1.64** | -12.56 | **-11.85** |
| | PATCHTST | 38.59 | **45.80** | 25.69 | **24.76** | 1.50 | **1.85** | -13.37 | **-12.48** |
| | iTRANSFORMER | 34.95 | **37.68** | **24.47** | 24.82 | 1.43 | **1.52** | -12.70 | **-11.56** |
| | MAMBA | 38.19 | **46.46** | 24.72 | **24.65** | 1.55 | **1.88** | -12.17 | **-12.10** |
| | BI-MAMBA+ | **39.40** | 39.17 | 25.96 | **24.13** | 1.52 | **1.62** | -12.28 | **-11.88** |
| | MODERNTCN | 29.56 | **33.88** | 24.73 | **24.21** | 1.20 | **1.40** | -13.24 | **-12.67** |
| | TCN | 33.14 | **40.22** | 24.85 | **24.85** | 1.33 | **1.62** | -13.88 | **-11.67** |

## F. Efficiency Analysis

In this section, we provide an empirical efficiency analysis of the proposed bi-level optimization framework. Following the main experimental setup, we benchmark the per-epoch training time of a standard predictive model against its bi-level counterpart under Scenario 1 on the CSI 1000 universe, using a single NVIDIA H20 GPU.

We choose CSI 1000 for this analysis because it is the largest universe considered in our experiments, thus presenting the heaviest computational load. Consequently, any additional overhead introduced by the bi-level procedure would be most evident in this setting.

We report results for a diverse set of sequence and time-series architectures. For each model, we measure the time required to complete a single training epoch under:

1. **Standard training**: conventional supervised learning without bi-level optimization.

2. **Bi-level training**: our proposed method with a single-step inner loop update.

The measured per-epoch training times on the CSI 1000 dataset are summarized in Table 9. The results show that, with a single inner-loop step, the bi-level method adds a moderate amount of computational overhead compared with standard training, and remains practically implementable across all tested architectures. This is particularly natural in quantitative finance, where the signal-to-noise ratio is typically low and models tend to use relatively modest parameter sizes to control overfitting. Under such model sizes, the extra computation required by the inner update is contained, and the overall training cost stays in a comparable range to that of standard supervised training.

We note that in other application domains with substantially larger models or more complex architectures, the relative overhead of bi-level optimization could be higher. However, within our forecasting task and model configurations, the impact on efficiency is not significant. At the same time, the bi-level approach effectively avoids the need for extensive repeated training runs for horizon selection, which would otherwise involve training tens or hundreds of separate models, leading to much higher total computational cost.

It is also important to emphasize that the absolute per-epoch times across different models in Table 9 are not meant to be directly compared as indicators of model quality or efficiency. The models have different architectures and parameter counts: for instance, Transformer-based models are, in principle, more computationally intensive than RNN-based models. Yet in practice, Transformer models in financial forecasting often need to be kept relatively small to mitigate overfitting, which can narrow the gap in actual runtime compared with lighter architectures. Therefore, the primary takeaway from this analysis is the relative overhead of bi-level training versus standard training for each given model, rather than cross-model runtime comparisons.

*Table 9.* **Comparison of Running Time.** Per-epoch training time (in seconds) on CSI 1000 under Scenario 1 using a single NVIDIA H20 GPU. Standard Training denotes conventional supervised training without bi-level optimization, while Bi-level Training denotes our method with a single inner-loop update per outer iteration. CSI 1000 is chosen as it corresponds to the largest universe and thus the heaviest computational load.

| Model | Standard Training (s/epoch) | Bi-level Training (s/epoch) |
|---|---|---|
| LSTM | 21.739 | 22.137 |
| GRU | 21.825 | 22.536 |
| DLinear | 21.568 | 22.672 |
| RLinear | 21.357 | 22.912 |
| PatchTST | 21.653 | 23.357 |
| iTransformer | 21.153 | 23.540 |
| Mamba | 21.912 | 25.688 |
| Bi-Mamba+ | 21.401 | 27.765 |
| ModernTCN | 21.317 | 26.583 |
| TCN | 21.484 | 23.284 |

