# OpenReview forum: "The Label Horizon Paradox: Rethinking Supervision Targets in Financial Forecasting"
_ICML.cc/2026/Conference — ICML 2026 regular_

### Official Review · Reviewer_8aQE · 2026-02-19

**Soundness:** 3
**Presentation:** 3
**Significance:** 3
**Originality:** 3
**Overall Recommendation:** 3
**Confidence:** 1

**Summary:**

This manuscript attempts to outline an important concept that the training supervision horizon in financial forecasting need not coincide with the inference horizon. The authors study an important area, short-term financial prediction under extremely low signal-to-noise regimes, where subtle modeling choices can materially affect generalization

**Compliance With Llm Reviewing Policy:**

Affirmed.

**Final Justification:**

Given the rebuttals and other reviews I would like to keep my score.

**Key Questions For Authors:**

Please see weakness above.

**Limitations:**

Is the method proposed in the paper a general supervision principle that could be used in any market or a phenomenon specific to intraday equity microstructure?

**Strengths And Weaknesses:**

* **Strength**:

1. The paper does a proper conceptual re-framing of supervision design. The central idea which is the Label Horizon Paradox, is intellectually compelling. The paper challenges a largely unquestioned convention: aligning the training label exactly with the evaluation target.

2. Theoretical and empirical aspects align cleanly. Many financial ML papers claim theoretical grounding. Here the derivation actually matches observed behavior.

3. The empirical evaluation across architectures are robust. The empirical section is broad and extensive.

4. The paper is clearly written and is technically sound.

* **Weakness**
1. There is no validation on  US and EU markets, asset classes like future and crypto, longer horizon thus, it remains unclear whether this is a general supervision principle or a phenomenon specific to intraday equity microstructure.
2. Bi-Level Optimization is methodologically incremental and not novel. While the application is novel (label selection instead of sample reweighting), the mechanism itself is standard.

---

> ### Author Rebuttal · Authors · 2026-03-27
>
> We sincerely thank the reviewer for finding our central idea intellectually compelling, praising the strong alignment between our theory and empirical results, and acknowledging the robustness of our evaluations.  We address your specific points below.
>
> ### 1. Generalization to US Markets (S&P 500)
> > **Reviewer's Concern:** *There is no validation on US market... thus, it remains unclear whether this is a general supervision principle...*
>
> **Response:**
> To confirm our framework is a **general supervision principle**, we conducted new experiments on the **US S&P 500** (identical daily prediction settings in the main text).
>
> | Model | $\mbox{IC}^{Std}$ | $\mbox{IC}^{Ours}$ | $\mbox{ICIR}^{Std}$ | $\mbox{ICIR}^{Ours}$ | $\mbox{RankIC}^{Std}$ | $\mbox{RankIC}^{Ours}$ | $\mbox{RankIR}^{Std}$ | $\mbox{RankIR}^{Ours}$ | $\mbox{TopRet}^{Std}$ | $\mbox{TopRet}^{Ours}$ | $\mbox{Sharpe}^{Std}$ | $\mbox{Sharpe}^{Ours}$ |
> |---|:---:|:---:|:---:|:---:|:---:|:---:|:---:|:---:|:---:|:---:|:---:|:---:|
> | LSTM | 0.314 | **0.459** | 0.302 | **0.441** | 0.149 | **0.219** | 0.139 | **0.206** | 0.141 | **0.159** | 1.443 | **1.600** |
> | GRU | 0.346 | **0.399** | 0.333 | **0.351** | 0.181 | **0.202** | 0.181 | **0.184** | **0.164** | 0.157 | 1.560 | **1.618**|
> | DLinearTS | 0.326 | **0.506** | 0.332 | **0.472** | 0.139 | **0.213** | 0.159 | **0.215** | 0.125 | **0.166** | 1.368 | **1.733** |
> | RLinearTS | 0.352 | **0.461** | 0.335 | **0.452** | 0.170 | **0.199** | 0.171 | **0.214** | 0.155 | **0.159** | 1.544 | **1.614** |
> | PatchTST | 0.309 | **0.489** | 0.347 | **0.421** | **0.180** | 0.178 | 0.169 | **0.216** | 0.137 | **0.148** | 1.330 | **1.570** |
> | iTransformer | 0.397 | **0.499** | 0.402 | **0.435** | 0.215 | **0.228** | 0.187| **0.232** | **0.177** | 0.175 | 1.712 | **2.097** |
> | Mamba | 0.429 | **0.527** | 0.406 | **0.539** | 0.192 | **0.241** | 0.180 | **0.245** | 0.160 | **0.175** | 1.640 | **1.875** |
> | Bi-Mamba+ | 0.315 | **0.473** | 0.236 | **0.417** | 0.177 | **0.225** | 0.124 | **0.182** | 0.149 | **0.164** | 1.409 | **1.653** |
> | ModernTCN | 0.306 | **0.455** | 0.247 | **0.383** | 0.141 | **0.198** | 0.111 | **0.171** | 0.136 | **0.155** | 1.231 | **1.486** |
> | TCN | 0.423 | **0.540** | 0.388 | **0.515** | 0.200 | **0.250** | 0.210 | **0.227** | 0.164 | **0.168** | 1.677 | **1.686** |
>
> As shown, our framework consistently outperforms the standard baseline. Additionally, we have **supplemented results under extreme market conditions in our response to Reviewer A8Bm**. This proves the Label Horizon Paradox is not a localized phenomenon.
>
> Furthermore, our theory mathematically guarantees this universality. The paradox is driven by a fundamental statistical trade-off: the competition between the Signal Growth Rate  and the Noise Accumulation. As long as an asset class exhibits delayed information diffusion and cumulative stochastic noise, this paradox will inevitably exist. Thus, our method remains universally applicable.
>
> ### 2. Novelty of the Bi-Level Optimization (BLO) Method
> > **Reviewer's Concern:** *BLO is incremental... While the application is novel, the mechanism itself is standard.*
>
> **Response:**
> We agree that the mathematical solver is a standard tool. However, we respectfully highlight that the fundamental contribution of this paper is the **discovery, theoretical explanation, and empirical validation of the Label Horizon Paradox**. Building upon this theoretical foundation, our methodological contribution lies not in inventing a new optimization solver, but in a novel Problem Formulation designed to directly address this newly identified challenge. Furthermore, we propose essential algorithmic enhancements tailored for high-noise environments.
>
> *   **Conceptual Innovation:** For years, the ML finance community has blindly accepted the dogma of strictly aligning training labels with evaluation targets. Recognizing this flaw and formulating the *label horizon* as a learnable meta-parameter is a fundamental shift from *data-centric* (e.g., sample reweighting) to *label-centric* learning. In the ML community, successfully applying a robust tool to solve a previously unexamined, high-impact problem constitutes a significant contribution.
> *   **Domain-Specific Adaptations:** Furthermore, applying standard BLO out-of-the-box to financial data (which has extremely low signal-to-noise ratios) fails due to meta-optimization collapse. To make this work, we introduced crucial domain-specific designs (Section 4.2): the **Standardized Mean-Field Warm-up** (to prevent shortcut learning on short-horizon labels, as ablated in Figure 4) and **Entropy Regularization** (to prevent weight collapse). These represent non-trivial methodological adaptations tailored specifically for high noise time series forecasting.
>
> We hope these new US market results and clarifications alleviate your concerns regarding generalizability and methodological contribution. We would be deeply grateful if you might consider raising your score.

---

> > ### Author Rebuttal · Reviewer_8aQE · 2026-04-02
> >
> > Thank authors for the rebuttal. I will keep my score.

---

> > > ### Author Response · Authors · 2026-04-06
> > >
> > > Thank you for your time, your engagement during the rebuttal.

---

### Official Review · Reviewer_A8Bm · 2026-03-06

**Soundness:** 3
**Presentation:** 3
**Significance:** 3
**Originality:** 4
**Overall Recommendation:** 4
**Confidence:** 4

**Summary:**

This paper highlights that in financial forecasting, optimizing the training label, not just the model or data, can be crucial, and the Label Horizon Paradox offers an in-depth label-centric perspective.

**Compliance With Llm Reviewing Policy:**

Affirmed.

**Final Justification:**

The authors have addressed all my concerns. These results have strengthened my assessment of the work, and as a result, I have raised my confidence score.

**Key Questions For Authors:**

See weaknesses.

**Limitations:**

Yes.

**Strengths And Weaknesses:**

Strengths:
1. The paper investigates a largely overlooked issue: whether the training label itself is optimal. This label-centric perspective offers an insightful angle for financial forecasting research.
2. The paper introduces the Label Horizon Paradox and argues that minimizing the training error for $t+\Delta$ does not necessarily lead to the best generalization at the same horizon.
3. The proposed framework provides a systematic way to identify an appropriate supervision horizon.
4. Extensive experiments on large-scale financial datasets are conducted, which demonstrate that the signal–noise trade-off indeed exists in real-world markets.

Weaknesses:
1. The core idea may be closely related to *label smoothing*, which could limit the technical contribution. The paper would benefit from clarifying the differences between the proposed method and label smoothing.
2. The theoretical explanation focuses on signal realization vs. noise accumulation, but real markets are influenced by many additional factors, such as liquidity conditions and unexpected news events. A deeper discussion of these factors would strengthen the paper.
3. The practical value of the method would be more convincing if supported by trading backtests demonstrating profitability.
4. Beyond reporting the Sharpe ratio, the paper could include stress tests (e.g., performance during market crashes) to evaluate the robustness of the method. In real trading scenarios, risk resilience can be as important as profitability.

---

> ### Author Rebuttal · Authors · 2026-03-28
>
> We sincerely thank the reviewer and are highly encouraged that you find our label-centric perspective and the Label Horizon Paradox insightful. Below, we address your constructive feedback.
>
> ### 1. Stress Tests During Market Crashes
>
> > **Reviewer's Concern:** *... the paper could include stress tests (e.g., performance during market crashes) ...*
>
> **Response:**
> To evaluate risk resilience, we conducted a new stress-test on the **CSI 500 index (Jul 2023–Jul 2024)**, capturing a severe downtrend and the **Feb 2024 market crash**.  Experimental settings match Scenario 1.
>
> | Model | $\mbox{IC}^{Std}$ | $\mbox{IC}^{Ours}$ | $\mbox{ICIR}^{Std}$ | $\mbox{ICIR}^{Ours}$ | $\mbox{RankIC}^{Std}$ | $\mbox{RankIC}^{Ours}$ | $\mbox{RankICIR}^{Std}$ | $\mbox{RankICIR}^{Ours}$ | $\mbox{TopRet}^{Std}$ | $\mbox{TopRet}^{Ours}$ | $\mbox{Sharpe}^{Std}$ | $\mbox{Sharpe}^{Ours}$ |
> |---|---:|---:|---:|---:|---:|---:|---:|---:|---:|---:|---:|---:|
> | LSTM | 0.874 | **0.909** | **0.927** | 0.916 | 0.817 | **0.932** | 1.013 | **1.104** | 0.129 | **0.131** | 1.411 | **1.454** |
> | GRU | 0.883 | **0.890** |0.887  | **0.928** | 0.843 | **0.926** | 1.016 | **1.053** | 0.123| **0.129**  | **1.414** | 1.341 |
> | DLinearTS | 0.746 | **0.817** | 0.801 | **0.817** | 0.749 | **0.865** | 0.914 | **1.015** | 0.103 | **0.125** | 1.164 | **1.362** |
> | RLinearTS | 0.879 | **0.884** | 0.921|  **0.979** | 0.897 | **0.924** | **1.117** | 1.086 | **0.130** | 0.126 | 1.367 |  **1.458**|
> | PatchTST | 0.884 | **0.931** | 0.857 | **0.861** | 0.862 | **0.974** | 0.928 | **1.005** | 0.128 | **0.136** | 1.387 | **1.469** |
> | iTransformer | 0.846 | **0.910** | 0.828 | **0.868** | 0.871 | **0.946** | 0.985 | **1.032** | 0.106 | **0.129** | 1.176 | **1.438** |
> | Mamba | 0.922 | **0.965** | 0.927 | **0.982** | 0.915 | **1.011** | 1.055 | **1.083** | 0.126 | **0.139** | 1.402 | **1.511** |
> | Bi-Mamba+ | 0.953 | **0.957** | **0.930** | 0.906 | 0.896 | **0.990** | 0.977 | **1.069** | 0.138  | **0.141**| 1.495 |  **1.542** |
> | ModernTCN | 0.740 | **0.847** | 0.744 | **0.823** | 0.698 | **0.868** | 0.762 | **0.929** | 0.089 | **0.119** | 0.981 | **1.319** |
> | TCN | 0.935 |  **0.951**| 0.940  | **0.978**| 0.929 | **0.964** | 1.049 | **1.097** | 0.129 | **0.134** | 1.397 | **1.451** |
>
> As shown in the table, even under extreme market stress, our framework also consistently outperforms the standard baselines.
>
> ### 2. Differences from Label Smoothing
> > **Reviewer's Concern:** *The idea may be related to label smoothing... clarify the differences.*
>
> **Response:**
> Although both methods manipulate labels, their mechanisms are fundamentally different:
> - **Standard Label Smoothing**: smooths the numerical distribution of a fixed target (e.g., [1, 0] → [0.9, 0.1]) as a regularization technique.
> - **Our method**: does not change the label distribution; it searches along the time axis and selects the optimal horizon as the training label.
>
> If your concern about label smoothing refers to mixing labels across horizons, we clarify that this differs from our horizon selection approach, and our experiments in Sec. 6.1 show it is less effective.
>
> ### 3. Other Influencing Factors
> > **Concern:** *…real markets are influenced by many factors… A deeper discussion would strengthen the paper*
>
> **Response:** We deeply appreciate this insightful comment and our response is grounded in two key aspects:
> * **Data-centric vs. Label-centric:** Identifying specific drivers like news or liquidity is a *data-centric* task. Our paper pioneers a strictly *label-centric* perspective, focusing on the mathematical evolution of the Signal-to-Noise Ratio over time rather than specific causal events.
> * **Implicit Inclusion in Theory:** These real-world factors are intrinsically captured in our framework. Liquidity dictates the rate of signal realization, while unexpected news acts as stochastic shocks absorbed into the noise accumulation term. Since quantitative investing relies on a statistical edge over many trades, our framework remains robust and practically valuable even if extreme black-swan events temporarily disrupt these assumptions.
>
> ### 4. Trading Backtests
> > **Reviewer's Concern:** *...would be more convincing if supported by trading backtests...*
>
> **Response:**
> While our simplified top-layer daily Return&Sharpe provide a reasonable profitability proxy, we acknowledge the omission of an industry-grade backtest. Realistic backtest results depend heavily on downstream modules (e.g., risk optimization, portfolio execution). Without them, profitability estimates can be misleading; yet, incorporating them would introduce confounding variables and dilute our core focus on the ML training paradigm.
>
> Furthermore, since real-world quantitative firms often decouple Alpha research from portfolio construction, isolating and demonstrating a improvement in pure model performance is also a valuable standalone contribution. We hope the reviewer agrees this focused evaluation remains a meaningful contribution.

---

> > ### Author Rebuttal · Reviewer_A8Bm · 2026-04-01
> >
> > I thank the authors for addressing my concerns.
> > I appreciate the clarifications about label smoothing,  discussions on other factors, and the thorough stress test analysis.
> >
> > For **W3**, I would like to note that many prior works still adopt simple and standardized backtesting strategies, e.g., the top-k strategy [1].
> > As long as the same strategy is consistently applied across all compared methods, such evaluations can still make the empirical results more convincing, even if they do not fully reflect real-world trading strategies.
> > I understand that the primary focus of this work is not on downstream trading or portfolio optimization.
> > Nevertheless, I would encourage the authors to consider including at least a simplified demonstration in future versions to further support the application values of the method.
> > Overall, my concerns are largely addressed, and I will keep my positive recommendation.
> >
> > [1] Li et al., R&D-Agent-Quant: A Multi-Agent Framework for Data-Centric Factors and Model Joint Optimization. NeurIPS, 2025.

---

> > > ### Author Response · Authors · 2026-04-02
> > >
> > > Thank you for your continued support, your positive recommendation, and for acknowledging our efforts in addressing your previous concerns.
> > >
> > > We deeply appreciate your constructive suggestion regarding the downstream backtesting. We completely agree that demonstrating a standardized trading strategy, as seen in [1], significantly strengthens the empirical evidence and application value of our method. Inspired by your advice, we have retrained the model and conducted an additional downstream backtesting experiment.
> > >
> > > **Experimental Setup:**
> > > Following the top-k strategy paradigm, we constructed a daily rebalanced, equal-weight portfolio. The detailed settings are as follows:
> > > * **Stock Universe & Period:** Constituents of the CSI 500, backtested from July 2024 to July 2025.
> > > * **Execution Strategy:** Models generate predictions 15 minutes before the market close. We then use the Time-Weighted Average Price (TWAP) of the subsequent 10 minutes as the execution price.
> > > * **Portfolio Construction:** Each day, we select and equally weight the top 20% of stocks with the highest predicted scores.
> > > * **Real-world Constraints:** Transaction costs and slippage are strictly incorporated into the backtesting framework to reflect real-world trading conditions.
> > > | Model | Ann. Ret(Std) | Ann. Ret(Our) | Ann. Vol(Std) | Ann. Vol(Our) | Sharpe(Std) | Sharpe(Our) | MDD(Std) | MDD(Our) |
> > > |---|---:|---:|---:|---:|---:|---:|---:|---:|
> > > | LSTM | 42.18% | **42.27%** | 25.89% | **24.53%** | 1.63 | **1.72** | -13.96% | **-11.25%** |
> > > | GRU |36.15% |  **40.90%** |  25.01%| **24.68%** |  1.45| **1.66** | -13.55%  | **-12.17%**|
> > > | DLinearTS | 34.41% | **41.65%** | 25.11% | **24.80%** | 1.37 | **1.68** | -12.84% | **-12.01%** |
> > > | RLinearTS | 33.97% | **41.04%** | **23.87%** | 24.95% | 1.42 | **1.64** | -12.56% | **-11.85%** |
> > > | PatchTST | 38.59% | **45.80%** | 25.69% | **24.76%** | 1.50 | **1.85** | -13.37% | **-12.48%** |
> > > | iTransformer | 34.95% | **37.68%** | **24.47%** | 24.82% | 1.43 | **1.52** | -12.70% | **-11.56%** |
> > > | Mamba | 38.19% | **46.46%** | 24.72% | **24.65%** | 1.55 | **1.88** | -12.17% | **-12.10%** |
> > > | Bi-Mamba+ | **39.40%** | 39.17% | 25.96% | **24.13%** | 1.52 | **1.62** | -12.28% | **-11.88%** |
> > > | ModernTCN | 29.56% | **33.88%** | 24.73% | **24.21%** | 1.20 | **1.40** | -13.24% | **-12.67%** |
> > > | TCN | 33.14% | **40.22%** | **24.85%** | **24.85%** | 1.33 | **1.62** | -13.88% | **-11.67%** |
> > >
> > > As demonstrated in the results, applying our method consistently improves downstream trading performance across almost all backbone architectures.
> > >
> > > Once again, we sincerely thank you for your time, effort, and constructive feedback, which have greatly improved the quality of our paper. If our additional experiments and responses have further strengthened your confidence in our work, we would be extremely grateful if you could consider raising your score.

---

### Official Review · Reviewer_b59B · 2026-03-08

**Soundness:** 2
**Presentation:** 3
**Significance:** 2
**Originality:** 3
**Overall Recommendation:** 4
**Confidence:** 3

**Summary:**

This paper observes that in traditional financial market forecasting tasks, due to the inherent complexity and high noise of the market, the supervision signal is often inaccurate, thereby limiting the predictive performance of related models. Through experimental data observation and theoretical logical reasoning, it summarizes the Label Horizon Paradox, revealing that the prediction horizon in return forecasting tasks is not necessarily the most suitable label for training. Subsequently, the paper proposes the Adaptive Horizon Learning via Bi-level Optimization method, which utilizes a bi-level structure to optimize the parameters of the backbone model and identify the optimal training supervision horizon. Tested on large-scale datasets, the experiments demonstrate that this framework can be adapted to various existing methods and achieve positive results.

**Compliance With Llm Reviewing Policy:**

Affirmed.

**Final Justification:**

After reading the authors' rebuttal, I change my overall rating to weak positive.

**Key Questions For Authors:**

1. The decomposition of stock price trends in the paper relies excessively on idealized assumptions, such as the monotonic increment of market information digestion and the observability of factor exposures.

2. During the inner loop, the model first trains the function using intermediate window proxy labels—whether these labels introduce data snooping bias remains unaddressed.

3. Why does the experimental section only present results of the scenarios 1, with a lack of real results from other scenarios?

**Limitations:**

There is a lack of more in-depth experimental validations, such as applying some SOTA time-series stock price forecasting models as the backbone; the generalization of the model lacks sufficient justification.

**Strengths And Weaknesses:**

Strengths：

1. This paper identifies a prevalent phenomenon in financial markets: labels of future returns are heavily contaminated by noise. Such noise restricts model performance when used as training labels. Through experiments, the paper summarizes the underlying patterns and proposes the "Label Horizon Paradox," which holds significant practical value.
2. The paper presents a highly extensible framework that can be adapted to existing advanced time-series forecasting methods to find more suitable label horizons for training, making it very valuable for practical applications.
3. Experiments validate the effectiveness of the bi-level optimization framework on three large-scale datasets and compare it against 10 categories of mainstream time-series models. Performance improvements are achieved across all models, demonstrating the method’s generalizability.

Weaknesses:

1. The technical design of the method is somewhat overly simplistic.
2. During the inner loop, the model first trains using intermediate window proxy labels—all derived from real prices, which are not utilized in traditional methods. This raises the possibility that the framework’s superior performance over traditional methods stems solely from these additional labels. However, sufficient validations are lacking to mitigate data snooping bias and randomness introduced by these labels.
3. The experiments focus on daily stock price forecasting. Daily stock price fluctuations typically exhibit universal patterns around key nodes such as market openings. It is possible that the model merely learns these universal patterns rather than the intrinsic nature of information digestion and pricing. Experimental results with other window lengths are absent to verify this conjecture.

---

> ### Author Rebuttal · Authors · 2026-03-27
>
> We sincerely thank the reviewer for recognizing the practical value of the Label Horizon Paradox and the comprehensiveness of our large-scale experiments. Below, we address your concerns point by point.
>
> ### 1. Data Snooping and the Role of Intermediate Labels
> > **Reviewer's Concern:** *the framework’s superior performance may stem solely from these additional labels (intermediate labels). Validations are lacking to mitigate data snooping bias.*
>
> **Response:**
> We deeply appreciate your rigorous check on data leakage. We kindly clarify that our method strictly prevents data snooping:
>
> *   **Strict Data Splits:** As detailed in **Appendix A**, the data splitting for the bi-level optimization is performed strictly and exclusively within the Training Set. The intermediate proxy labels are never exposed during inference. On the Test Set, the model strictly uses historical inputs to predict the final target without any future information leakage.
> *   **Beyond "Additional Labels":** In our experiments (e.g., Table 1), our final model is trained using **only a single specific horizon label** (identified by our framework). This is directly compared against the traditional baseline, which also uses exactly one label (the final target). Thus, both methods construct the training label from a single timestamp, ensuring a perfectly fair comparison. **Fundamentally, our work's essence is pinpointing this single optimal label, driven by our theoretical finding: the signal-to-noise ratio exhibits a systematic pattern over time.** Furthermore, **simply aggregating all intermediate labels is less effective**. As demonstrated in Section 6.1 and Appendix E, Naive Averaging or Multi-Task Learning  across all proxy labels performs worse than our method.
>
> ### 2. Generalization to Other Scenarios
> > **Reviewer's Concern:** *The model might merely learn universal patterns around market openings. Results with other scenario are absent.*
>
> **Response:**
> To ensure fair comparison with standard daily-prediction setup, we focused on Scenario 1 in the main text, deferring comprehensive results to the Appendix.
>
> *   **Supplementary Results:** **Appendix E (Tables 3 & 4)** provides full experimental results for **Scenario 2** and **Scenario 3**.  As expected, our method matches the baseline when the optimal and final labels align (Scenario 2), and clearly outperforms it when they deviate (Scenario 3), which perfectly aligns with the empirical brute-force search results in Figure 2. **Furthermore**, as detailed in our responses to other reviewers, we have newly validated the generalization of our framework across extended multi-day inputs, the U.S. S&P 500, and extreme stress tests.
> *   **Market Opening Conjecture:** Scenario 3 focuses on mid-day trading (far from the market open). As shown in **Figure 2c**, we still observe a distinct **hump-shaped performance curve**. Our framework successfully captures this intermediate optimal horizon, yielding significantly higher performance than the baseline (Table 4).
>
> ### 3. Justification of Theoretical Assumptions
> > **Reviewer's Concern:** *...relies on idealized assumptions...*
>
> **Response:**
> While assumptions like Arbitrage Pricing Theory (APT) and Gaussian noise may seem idealized, they are extensively validated, foundational tools in quantitative finance. They elegantly capture the core dynamic of the problem.
>
> The true value of our framework is its empirical alignment. As shown in Corollary 3.1 and Figure 3, our theoretical curve perfectly matches real-world performance dynamics. Furthermore, Appendix D provides an alternative proof using the purely statistical Partial Correlation Formula, reaching the exact same decomposition identity **without relying on the APT model**.
>
> ### 4. Simplicity of Technical Design and SOTA Model
> > **Reviewer's Concern:** *The technical design is simplistic...Lack of SOTA models as the backbone.*
>
> **Response:**
> *  **SOTA Model:** As listed in Section 5.2, we actually included recent SOTAs (e.g., PatchTST, iTransformer, Bi-Mamba+) as backbone. However, rather than merely chasing the absolute SOTA, our primary goal is to demonstrate the framework’s universal applicability. Therefore, we deliberately prioritized representativeness by selecting two distinct models from each of the five major architectural categories.
> *  **Innovation and Efficiency over Simplicity:** Bi-level Optimization for dynamic label selection is highly novel in noisy financial environments. Standard LOB is notoriously unstable here; thus, we introduce a critical warm-up strategy. Crucially, this warm-up guarantees stable convergence, allowing us to approximate the inner loop with just **a single gradient step**. This renders our framework exceptionally efficient (Appendix G) and scalable to massive high-frequency datasets.
>
> We hope these clarifications and the detailed evidence in the Appendix address your concerns. We would be deeply grateful if you would consider raising your score.

---

> > ### Author Rebuttal · Reviewer_b59B · 2026-04-04
> >
> > Thanks for the replying, and most of my concers are addressed. I would like to adjust my scores accordingly.

---

> > > ### Author Response · Authors · 2026-04-06
> > >
> > > Thank you very much for your time, constructive feedback, and for adjusting your score. We are glad that our response has addressed your concerns.

---

### Official Review · Reviewer_F8Hh · 2026-03-11

**Soundness:** 4
**Presentation:** 3
**Significance:** 3
**Originality:** 3
**Overall Recommendation:** 5
**Confidence:** 4

**Summary:**

This paper provides an in-depth analysis of supervision targets in financial time-series forecasting. Both theoretical derivations and empirical results demonstrate that the optimal supervision signal often deviates from the actual prediction goal. In addition, the authors propose a bi-level optimization framework that dynamically integrates various supervision signals to enhance model performance.

**Compliance With Llm Reviewing Policy:**

Affirmed.

**Final Justification:**

Thanks to the authors for the additional experimental details and the new results. They adequately address my questions regarding the setup and different market conditions.

Regarding the metrics, while I understand that a divergence between predictive accuracy and profitability is not necessarily contradictory, I maintain a slightly different perspective on the emphasis placed on predictive signals. In the context of financial applications, a predictive signal’s value is ultimately realized through its ability to generate profit. Therefore, a fair and thorough comparison of profit-based metrics is essential to fully demonstrate a method's effectiveness.

However, this difference in perspective does not detract from the overall quality of the work. I find your explanation of the metric divergence to be reasonable and technically sound. I now consider my concerns resolved and will adjust my score accordingly.

**Key Questions For Authors:**

How does the proposed framework perform across different historical periods and diverse international markets?

**Limitations:**

Yes

**Strengths And Weaknesses:**

Strengths:

1.	The core idea and theoretical analysis are highly engaging. By balancing market signal realization against noise accumulation, the paper provides a compelling rationale for why shifting the supervision signal is beneficial. The trade-off between model "accuracy" and "alignment” offers a novel and practical lens for financial forecasting.
2.	The theoretical framework is comprehensive and well-grounded. The authors use established financial theories to derive the necessity of differentiating the training signal from the prediction goal. Furthermore, the analysis clearly defines the conditions and boundaries for the proposed theory.
3.	The proposed method is grounded in theory and avoids the inefficiency of brute-force grid searches for optimal signals. Instead, the introduction of a dynamic bi-level optimization framework allows the model to autonomously select and weight the most effective supervision signals during training.

Weaknesses:

While the conceptual and theoretical contributions are significant, the experimental validation is insufficient. Several theoretical claims lack robust empirical support, which weakens the paper's overall contribution. The primary deficiencies in the experiments are as follows:

1.	The diversity and representativeness of the experimental market are limited. All experiments are conducted exclusively on the CSI indices, with a total data span of only six years and a test set covering just one year. This introduces significant geographical and temporal limitations. Financial markets exhibit vast differences, and since the paper‘s thesis is closely tied to market efficiency and information diffusion, experiments based on a single market cannot sufficiently prove the theory's generalizability. Specifically, the exclusion of the U.S. market, the world’s largest and relatively most efficient, is a major oversight. A complete validation requires a discussion of different market conditions with varying levels of efficiency, noise, and information transmission rates. The current experimental results are too narrow.
2.	The input features are restricted to minute-level price information from a single day (or less). There are no experiments exploring different input lengths, such as incorporating data from multiple preceding days. This raises a fundamental question regarding the theoretical definition of "noise": is the observed noise truly stochastic, or is it merely effective information that the model failed to extract due to incomplete input data? As the authors note, "Alpha requires time to be absorbed by the market", a concept extensively studied in information diffusion theory. Real-world markets rarely price in all information instantaneously. Numerous studies show that the lag between market information and price reflection can last from three days to two weeks or even longer, even in developed markets [1][2][3]. Consequently, concluding that the test period signal contains too much noise to be a suitable supervision signal, based solely on one day of input data, is premature and lacks sufficient evidence.

In conclusion, I recognize the theoretical and methodological value of this work and believe its effectiveness is preliminarily validated. However, the inherent limitations of the experimental setup constrain the scalability and reliability of the proposed theory.

---

> ### Author Rebuttal · Authors · 2026-03-29
>
> We sincerely thank the reviewer for the Weak Accept recommendation and for praising our theoretical framework as "comprehensive and well-grounded". We deeply appreciate your insightful questions regarding the experimental scope, which we address below.
>
> ### 1. Market Diversity and Generalizability
> > **Concern:** *Experiments lack diversity in markets (e.g., no U.S. market) and historical periods...*
>
> **Response:** We fully agree that validating across diverse markets and time periods is essential. To directly address your concerns, we have conducted two new comprehensive experiments (under the same daily prediction setting as the main text) during this rebuttal phase:
>
> *   **Diverse Markets & Efficiencies (U.S. Market – S&P 500):** As detailed in our response to **Reviewer 8aQE**, we evaluated our framework on the **S&P 500** for daily predictions. Notably, models trained with our method consistently outperform those trained with the standard paradigm, demonstrating that our gains also hold in the U.S. market.
> Moreover, as you pointed out, the U.S. market is more efficient and thus absorbs information faster. From our theory, this implies a faster and more significant performance decay over the prediction horizon. We indeed observe this phenomenon empirically, supporting its generalizability to highly efficient markets.
> *   **Diverse Historical Periods (Market Crash Stress Test):** To address temporal diversity, we conducted a stress test on the **CSI 500 from July 2023 to July 2024**, a period characterized by a severe, continuous downtrend and extreme volatility (detailed in our response to **Reviewer A8Bm**). Even under such extreme market stress, our framework maintains improvements, proving its robustness across entirely different historical market regimes.
>
> ### 2. Input Length
> > **Reviewer's Concern:** *Input is restricted to 1 day...model failed to extract due to incomplete input data?*
>
> **Response:**
> This is a profound observation. We agree that longer input periods may capture more persistent signals, which could theoretically push the optimal supervision horizon ($\delta^*$) closer to the final evaluation horizon ($\Delta$).
>
> However, market movements are inherently driven by a complex mixture of factors. Even when models are equipped with extended historical inputs to extract persistent alpha, fast-decaying transient signals also affect the final pricing dynamics. Consequently, the optimal label horizon $\delta^*$ may continue to deviate from $\Delta$.
>
> To empirically validate that our method holds beyond 1-day inputs, we conducted a new daily prediction experiment on the CSI 500 using LSTM with extended input sequences.  **While maintaining the same daily prediction setting as the main text, we increased the patch size and reduced the sequence length to accommodate the longer historical inputs.**
>
> |Length | $\mbox{IC}^{Std}$ | $\mbox{IC}^{Ours}$ | $\mbox{ICIR}^{Std}$ | $\mbox{ICIR}^{Ours}$ | $\mbox{RankIC}^{Std}$ | $\mbox{RankIC}^{Ours}$ | $\mbox{RankICIR}^{Std}$ | $\mbox{RankICIR}^{Ours}$ | $\mbox{TopRet}^{Std}$ | $\mbox{TopRet}^{Ours}$ | $\mbox{Sharpe}^{Std}$ | $\mbox{Sharpe}^{Ours}$ |
> |---|---:|---:|---:|---:|---:|---:|---:|---:|---:|---:|---:|---:|
> | 1 days | 0.716 | **0.840** | 0.601 | **0.737** | 0.444 | **0.466** | 0.601 | **0.674** | 0.286 | **0.306** | 2.640 | **2.803** |
> | 2 days | 0.729 | **0.817** | 0.594 | **0.644** | 0.481 | **0.481** | 0.594 | **0.637** | 0.302 | **0.308** | 2.787 | **2.871** |
> | 3 days | 0.675 | **0.805** | 0.535 | **0.544** | 0.477 | **0.496** | 0.535 | **0.622** | 0.299 | **0.318** | 2.756 | **3.012** |
> | 4 days | 0.704 | **0.825** | 0.570 | **0.673** | 0.494 | **0.530** | 0.570 | **0.642** | 0.307 | **0.325** | 2.854 | **3.047** |
> | 5 days | 0.709 | **0.824** | 0.593 | **0.676** | 0.468 | **0.561** | 0.593 | **0.626** | 0.312 | **0.327** | 2.828 | **3.095** |
>
> As shown above, even with longer historical inputs, our method continues to significantly outperform traditional training (which strictly uses $\Delta$). This proves that the deviation of the optimal supervision signal persists across different input lengths.
>
> **Furthermore**, it is crucial to emphasize that both our theoretical framework and dynamic selection algorithm are **universally applicable**. Neither our theory nor our method forcefully assumes that the optimal label *must* deviate from the final target. In real-world applications, different forecasting scenarios utilize vastly different input windows, and our framework simply identifies the empirical optimum for *any given input*.
>
> If, in a certain scenario, extending the input makes the final horizon ($\Delta$) the true optimal target, our bi-level optimization will autonomously assign it the highest weight. This adaptive behavior is exactly what we demonstrated in **Scenario 2** of the main text, where our algorithm successfully identified the final evaluation horizon ($\Delta$) as the true optimal label.

---

> > ### Author Rebuttal · Reviewer_F8Hh · 2026-04-03
> >
> > Thank you for the rebuttal and the additional experiments. The new results are good,  but the lack of detailed documentation regarding the experimental setup makes them difficult to evaluate fully. I have several remaining concerns:
> >
> > There is currently no introduction for the newly added experiments. Specifically, regarding the US market and crash stress tests, please provide the following details. How was the data structured for these specific scenarios? What are the exact timeframes and sizes for the training, validation, and test sets?
> >
> > Additionally, the added market crash stress test does not fully address my previous concerns regarding diverse market conditions. The new test data is only one year apart from the original test period. This is insufficient to represent a fundamentally different market regime. To provide truly persuasive evidence of robustness, results from significantly different eras, such as the pre-2020 (COVID-19 onset) or pre-2010 (Global Financial Crisis) periods, can be more promising.
> >
> > The results for longer input sequences look good but present a puzzling trend: non-profit  metrics consistently decrease as input length increases, yet profit metrics consistently increase. While I acknowledge that predictive metrics and realized profit do not always share a linear relationship, this consistent contradiction is concerning. Does this indicate that while changing training signals may improve certain predictive signals, it does not translate into fundamental improvements in strategy profitability? If this is true, the effectiveness of the signal change is questionable. Please provide a deeper analysis of this phenomenon and explain the underlying mechanism driving these diverging trends.

---

> > > ### Author Response · Authors · 2026-04-06
> > >
> > > **1. Detailed Setup**
> > >
> > > Both new scenarios strictly follow the Scenario 1 (daily prediction) protocol from the main paper (Table 1), including data structure, feature engineering, and label construction, with the only adjustment being the shifted timeframes as detailed below:
> > > *   U.S. Market: Evaluated on the S&P 500 index, the dataset is split into Jan. 2019–Jul. 2023 (Train), Jul. 2023–Jul. 2024 (Val), and Jul. 2024–Jul. 2025. The only adjustment is a longer input sequence corresponding to the longer U.S. daily trading session in a day.
> > > *   A-share Stress Test: Evaluated on the CSI 500 index, the dataset is split into Jan. 2018–Jul. 2022 (Train), Jul. 2022–Jul. 2023 (Val), and Jul. 2023–Jul. 2024 (Test, covering the severe Feb. 2024 market crash).
> > >
> > > **2. Regarding Pre-2020 Market Conditions**
> > >
> > > We present the pre-2020 S&P 500 evaluation below to reflect earlier conditions. The configuration is identical to the U.S. setup above, with the timeframe adjusted to: Jan. 2014–Jan. 2018 (Train), Jan. 2018–Jan. 2019 (Val), and Jan. 2019–Jan. 2020 (Test).
> > >
> > > | Model | $\mbox{IC}^{Std}$ | $\mbox{IC}^{Ours}$ | $\mbox{ICIR}^{Std}$ | $\mbox{ICIR}^{Ours}$ | $\mbox{RankIC}^{Std}$ | $\mbox{RankIC}^{Ours}$ | $\mbox{RankICIR}^{Std}$ | $\mbox{RankICIR}^{Ours}$ | $\mbox{TopRet}^{Std}$ | $\mbox{TopRet}^{Ours}$ | $\mbox{Sharpe}^{Std}$ | $\mbox{Sharpe}^{Ours}$ |
> > > |---|---:|---:|---:|---:|---:|---:|---:|---:|---:|---:|---:|---:|
> > > | LSTM | 0.287 | **0.325** | 0.274 | **0.290** |  0.219 | **0.226**|0.240  |  **0.292**|  0.164|  **0.171**| 2.583  | **2.783**|
> > > | GRU | 0.312 | **0.327** | 0.288 | **0.315** | **0.243** | 0.241 |0.236 | **0.302**  | **0.180** | 0.172 | 2.692 |  **3.271**|
> > > | DLinearTS | 0.281 | **0.350** | 0.252 | **0.302** | 0.125 | **0.167** | 0.154 | **0.169** | 0.158 | **0.168** |  2.732|  **2.734**|
> > > | RLinearTS | 0.312 | **0.358** | **0.334** | 0.321 | 0.202 |  **0.246**|  0.214 | **0.311**| 0.176 | **0.183** |  2.859|  **3.078**|
> > > | PatchTST | 0.304 | **0.414** | 0.309 | **0.387** | 0.252 | **0.298** | 0.264 | **0.298** | 0.188 | **0.190** | 2.960  | **3.097**|
> > > | iTransformer | 0.348  | **0.359**|  0.301 | **0.383**| 0.172  | **0.236**|  0.152 | **0.297**| 0.172 | **0.179** | 2.919|  **3.150** |
> > > | Mamba | 0.338 | **0.381** | 0.339 | **0.352** | 0.228 | **0.251** | **0.284** | 0.272 | **0.187** | 0.181 |  2.956|  **3.371**|
> > > | Bi-Mamba+ | 0.217 | **0.365** | 0.213 | **0.330** | 0.250  | **0.253**|  0.237 | **0.290**| 0.161 | **0.175** |  2.681|  **2.716**|
> > > | ModernTCN |  0.278 |**0.324** | 0.193| **0.286**  |  0.153 | **0.289**|  0.201 | **0.313**|  0.167 | **0.182**|  2.458| **3.108** |
> > > | TCN | 0.350 | **0.382** | 0.324 | **0.331** | 0.239 | **0.243** | 0.242| **0.296**  |  0.166  | **0.185**| 2.721 |  **3.317**|
> > >
> > >
> > > **3. On the Diverging Trends Between Predictive Metrics and Profitability**
> > >
> > > We appreciate the reviewer's careful observation regarding the diverging trends . However, this divergence is not necessarily contradictory. As is widely recognized, profit metrics depend not only on the predictive signals (which are our primary focus) but also heavily on the downstream trading rules that convert these signals into actual positions.
> > >
> > > To ensure a rigorously align with established baselines, we adopt a rather naive, widely accepted equal-weigh top-k protocol (e.g., top 10%)—a standard evaluation method in academic papers (including many ICML works). The divergence arises precisely from the simplistic nature of this setting and what the metrics fundamentally capture: metrics like IC/RankIC measure the *global* correlation over the full cross-section of stocks, whereas a naive top-k strategy's PnL is driven exclusively by the upper tail, ignoring the relative signal strengths within this top group (because of the equal-weigh). Consequently, global predictive metrics can decrease even when the tail ranking—and thus profitability—improves. Here, the top-k protocol serves merely as a simplistic, standardized example. Different backtesting strategies might alter or eliminate this divergence, but this is secondary. How to further unleash the potential of better predictive signals through advanced portfolio optimization falls outside the scope of this paper.
> > >
> > > While aligned trends are intuitive, even metrics like IC and RankIC rarely move in perfect lockstep. Due to this inherent non-monotonicity, related literature commonly evaluates predictive signals using diverse metrics, although IC is typically the first metric of interest. **Hence, we report multiple metrics to guarantee a robust assessment of the signal's generality**, without claiming universal monotonicity.  **What truly matters is that under any identical configuration (same backbone, input length, scenario, and strategy), our method consistently outperforms baselines across these metrics. This proves its intrinsic value, whereas the variations and correlations of metrics across different settings are not the focus of this work.**

---

### Decision · Program_Chairs · 2026-04-30

**Decision:**

Accept (regular)

**Comment:**

This paper argues that, in financial forecasting, the best supervision horizon may differ from the inference horizon, and proposes a bi-level optimization framework to identify effective proxy labels during training. Reviewers generally agreed that this label-centric perspective is interesting and potentially important, and viewed the combination of theory and empirical evidence as a core strength. They also found the empirical study broad across architectures, with added evidence suggesting the phenomenon persists across different settings.

The main concerns were whether the empirical validation was broad enough to support strong generality claims, how much novelty lies in the optimization framework itself versus the problem formulation and analysis, and whether profitability-oriented evaluation should play a larger role in a financial forecasting paper. The rebuttal substantially addressed these concerns by adding evidence on U.S. market data, stress-test periods, longer input settings, and downstream trading results, while also clarifying the distinction between the proposed approach and simpler alternatives such as label smoothing or naive use of multiple labels.

Overall, while some limitations remain regarding the breadth of financial settings studied and the incremental nature of the optimization machinery, the paper makes a clear and well-supported contribution by identifying and validating an under-explored issue in supervision design for financial forecasting. On balance, I recommend acceptance.